

# Photon counting statistics in Gaussian bosonic networks

Kalle Sulo Ukko Kansanen[1,2⋆], Pedro Portugal[2],
Christian Flindt[2,3] and Peter Samuelsson[1]

**1** Department of Physics and NanoLund, Lund University, Sweden
**2** Department of Applied Physics, Aalto University, Finland
**3** RIKEN Center for Quantum Computing, Wakoshi, Saitama 351-0198, Japan

⋆ kalle.kansanen@gmail.com

## Abstract

The statistics of transmitted photons in microwave cavities play a foundational role in microwave quantum optics and its technological applications. By utilizing quantum mechanical phase-space methods, we here develop a general theory of the photon counting statistics in Gaussian bosonic networks consisting of driven cavities with beamsplitter interactions and two-mode-squeezing. The dynamics of the network can be captured by a Lyapunov equation for the covariance matrix of the cavity fields, which generalizes to a Riccati equation, when counting fields are included. By solving the Riccati equation, we obtain the statistics of emitted and absorbed photons as well as the time-dependent correlations encoded in waiting time distributions and second-order coherence functions. To illustrate our theoretical framework, we first apply it to a simple linear network consisting of two coupled cavities, for which we evaluate the photon cross-correlations and discuss connections between the photon emission statistics and the entanglement between the cavities. We then consider a bosonic circulator consisting of three coupled cavities, for which we investigate how a synthetic flux may affect the direction of the photon flow, similarly to recent experiments. Our general framework paves the way for systematic investigations of the photon counting statistics in Gaussian bosonic networks.



# 1 Introduction

Networks of coupled bosonic modes provide a unique platform for realizing exotic quantum many-body physics and for exploiting non-classical correlations in future quantum technologies. As an example, bosonic networks form the basis of continuous-variable quantum computing, which is currently being implemented in several experimental setups [1–7]. Bosonic networks based on circuit quantum electrodynamics operate in the microwave regime [8, 9], and they can be used to simulate vibrational shifts in molecular spectra [10], investigate the topological properties of bosonic Kitaev chains [11], as well as to generate multipartite entangled states [12]. Bosonic networks can in principle be realized with any bosonic modes in the quantum regime, including nanomechanical resonators that are cooled to sub-Kelvin temperatures [13–17]. The combination of external drives and the coherent interactions between the modes can give rise to a wide range of physical phenomena such as entanglement, squeezing, and quantum interference. Understanding and harnessing these processes are essential for exploring quantum many-body physics with bosons and for advancing bosonic quantum technologies.

In networks made of coupled microwave cavities, photons may be emitted or absorbed because of the interactions between the confined electromagnetic fields within the cavities and their environments. The transmitted photons are expected to carry information about the network and its dynamics, and experimental progress towards the accurate detection of individual microwave photons is currently being made [18–28]. Such microwave photon detectors would provide an alternative way to access the state of the network as compared to methods based on dispersive readout. In one type of detector, the absorption of a microwave photon enables the inelastic tunneling of an electron in a double quantum dot, whose charge state is monitored in real time [22–24, 29]. There are also bolometric and calorimetric schemes, in which the temperature of a mesoscopic reservoir abruptly changes as a microwave photon is absorbed [30–32]. The energy of a microwave photon is comparable to the temperature in sub-Kelvin experiments, such that thermal effects become important. For example, heat currents may be generated between the cavities and their environments, and those are carried both by the photons that are emitted and those that are absorbed. The heat carried by photons in microwave cavities [33] and in electrical circuits [34] has been investigated, and the statistics of heat fluctuations is now a central topic in quantum thermodynamics [35,36]. Theoretically, the photon counting statistics of single bosonic modes has been investigated [37–41]. By con-

trast, much less is known about the photon counting statistics of coupled microwave cavities, and systematic and general investigations have so far been lacking.

Here, we develop a theoretical framework for evaluating the photon counting statistics of coupled microwave cavities. As illustrated in Fig. 1, the cavities may be excited by external drives, and they can be coupled by beamsplitter interactions and two-mode-squeezing. The cavities also exchange photons with their environments through emission and absorption processes. The coupling to the environments is weak, and we describe the cavities by a Hamiltonian, which at most is quadratic in the bosonic creation and annihilation operators. We can then employ quantum mechanical phase-space methods, which enable a convenient description of the network. We also include counting fields that couple to the number of emitted and absorbed photons, which makes it possible to evaluate the photon counting statistics on all relevant time scales. At short times, we consider the time-dependent correlations as described by the distribution of waiting times [37–42] and second-order coherence functions [43]. At long times, we consider the cumulants of the photon currents and their large-deviation statistics [44]. While earlier works have considered the photon counting statistics of single microwave cavities [37–41], our general framework makes it possible to treat any number of coupled cavities. Here, it is worth mentioning an alternative approach based on the theory of third quantization [45–47], which has also been extended to full counting statistics [47]. As applications, we consider small bosonic networks with two or three coupled modes as realized in recent experiments [13–17]. We focus on the cross-correlations between the outgoing photons, the entanglement between the cavities, as well as on a bosonic circulator, where a synthetic flux is used to control the direction of the photon flow.

The rest of our article is organized as follows. In Sec. 2, we introduce the Hamiltonian of the Gaussian bosonic network and the quantum master equation, which accounts for the transfer of photons between the cavities and their environments. We then employ a phase-space representation, which allows us to describe the state of the bosonic network by a displacement vector and a covariance matrix only. In Sec. 3, we introduce counting fields that couple to the number of emitted and absorbed photons, which make it possible to evaluate the photon counting statistics. With counting fields included, the dynamical equation for the covariance matrix becomes a Riccati equation, which we can solve both at finite times and at long times. In Sec. 4, we illustrate our general theoretical framework with three specific applications. We first consider two coupled cavities for which we investigate the photon counting statistics with an emphasis on the cross-correlations between the emitted photons from the two cavities. Second, we discuss the entanglement between the cavities and how it may be detected from measurements of the photon counting statistics. As our last application, we consider a bosonic circulator, where a synthetic flux is used to control the flow of photons between three cavities. Finally, in Sec. 5, we conclude on our work and provide an outlook on possible avenues for further developments. Several technical details are deferred to the Appendices.

## 2 Gaussian bosonic networks

We consider a network of $N$ coupled bosonic modes as illustrated in Fig. 1. To base our discussion on a specific physical setting, we focus on resonant modes of microwave cavities. However, it should be clear that our formalism applies to any other bosonic modes, such as nanomechanical resonators in the quantum regime [13–17]. The cavities have the eigenfrequencies $\omega_j$ and are all coherently driven at the frequency $\omega_D$. We denote the annihilation and creation operators for mode $j$ $(= 1, \ldots, N)$ by $\hat{a}_j$ and $\hat{a}_j^\dagger$, which obey the canonical commutation relations $\left[\hat{a}_j, \hat{a}_k^\dagger\right] = \delta_{jk}$. To keep the discussion general, we consider a quadratic Hamiltonian that includes all possible pairwise interactions between the cavities, given in the

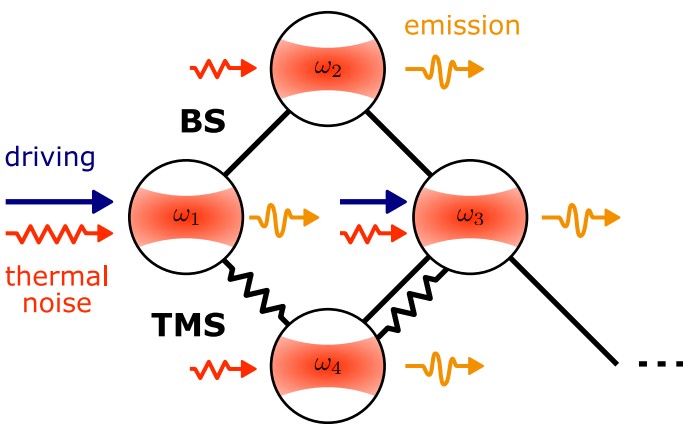

Figure 1: Gaussian bosonic network. The mode frequencies are denoted by $\omega_j$, and the modes can be coupled by beamsplitter (BS) interactions and two-mode-squeezing (TMS). The cavities emit (orange arrows) and absorb photons from the external drive (blue arrows) and the thermal reservoirs (red arrows).

rotating frame of the drive, as ($\hbar = 1$)

$$\hat{H} = \sum_{j=1}^{N} \left[ \delta\omega_j \left( \hat{a}_j^\dagger \hat{a}_j + \frac{1}{2} \right) + \frac{1}{2} \left( r_j \hat{a}_j^{\dagger 2} + r_j^* \hat{a}_j^2 \right) - i \left( f_j^* \hat{a}_j - f_j \hat{a}_j^\dagger \right) \right]$$
$$+ \sum_{k>j}^{N} \sum_{j=1}^{N} \left( g_{jk} \hat{a}_j^\dagger \hat{a}_k + g_{jk}^* \hat{a}_k^\dagger \hat{a}_j + \lambda_{jk} \hat{a}_j^\dagger \hat{a}_k^\dagger + \lambda_{jk}^* \hat{a}_j \hat{a}_k \right). \tag{1}$$

Here, the detuning from the eigenfrequency of cavity $j$ is denoted by $\delta\omega_j = \omega_j - \omega_D$, and $f_j$ is the amplitude of a coherent drive of mode $j$. The beamsplitter (BS) interaction, with strength $g_{jk}$, interchanges photons between two cavities, $j \neq k$, and conserves the total number of photons. By contrast, the two-mode-squeezing (TMS) interaction, with strength $\lambda_{jk}$, destroys (or creates) one photon in each cavity $j \neq k$, thereby changing the photon number by $\pm 2$. The single-mode squeezing interaction, with strength $r_j$, creates or destroys two photons in the same cavity $j$, arising from parametrically driving the cavity with a frequency close to $2\omega_j$. We note that single-mode squeezing [4, 48] as well as BS [10, 49] and TMS [50] interactions between bosonic modes have been implemented experimentally in a variety of physical systems. It should also be mentioned that with several different coherent and parametric drive frequencies, the Hamiltonian would be time-dependent, which we will not consider here. This choice implies that the cavity eigenfrequencies have to be the same in the presence of driving.

To be concise, we introduce a matrix notation. To this end, we collect the bosonic operators in a column vector $\hat{\boldsymbol{a}} = (\hat{a}_1, \hat{a}_1^\dagger, \ldots, \hat{a}_N, \hat{a}_N^\dagger)^T$, and introduce a vector of drive amplitudes $\boldsymbol{f} = (f_1, f_1^*, \ldots, f_N, f_N^*)^T$ as well as the diagonal matrix $\mathcal{K} = \text{diag}(1, -1, \ldots, 1, -1)$. The Hamiltonian in Eq. (1) can then be written as

$$\hat{H} = \frac{1}{2} \hat{\boldsymbol{a}}^\dagger \mathcal{H} \hat{\boldsymbol{a}} - i \boldsymbol{f}^\dagger \mathcal{K} \hat{\boldsymbol{a}}, \tag{2}$$

in terms of a $2N \times 2N$ Hermitian matrix $\mathcal{H}$, which contains all parameters of the Hamiltonian except those of the coherent drive, which are contained in the vector $\boldsymbol{f}$.

The cavities are weakly coupled to separate and independent thermal reservoirs at temperature $T_j$. In contrast to optical cavities, thermal effects are relevant in the microwave regime, with the equilibrium populations of the cavity modes being finite. The weak coupling to the reservoirs gives rise to the dissipation rates $\gamma_j \ll \omega_j$. Here, we neglect pure dephasing and

losses to any other environments than the reservoirs. The cavity network is therefore a dissipative and potentially driven system, whose density matrix $\hat{\rho}(t)$ evolves according to the quantum master equation [51–53]

$$\frac{\mathrm{d}}{\mathrm{d}t}\hat{\rho}(t) = \mathcal{L}\hat{\rho}(t) = -i\big[\hat{H},\hat{\rho}(t)\big] + \sum_{j=1}^{N}\gamma_j\mathcal{D}_j\hat{\rho}(t). \tag{3}$$

Here, the Liouvillian $\mathcal{L}$ generates the dynamics of the network including the unitary evolution and the incoherent dynamics caused by photon emissions to and from cavity $j$ with the bare rate $\gamma_j$. These incoherent processes are described by the Lindblad superoperators

$$\mathcal{D}_j\hat{\rho} = \bar{n}_j\left(\hat{a}_j^\dagger\hat{\rho}\hat{a}_j - \frac{1}{2}\big\{\hat{\rho},\hat{a}_j\hat{a}_j^\dagger\big\}\right) + (\bar{n}_j+1)\left(\hat{a}_j\hat{\rho}\hat{a}_j^\dagger - \frac{1}{2}\big\{\hat{\rho},\hat{a}_j^\dagger a_j\big\}\right), \tag{4}$$

where $\bar{n}_j = 1/\big(\exp\big[\omega_j/k_B T_j\big]-1\big)$ is the Bose–Einstein distribution at the cavity frequency $\omega_j$.

Since our goal is to describe photon emissions and absorptions from specific cavities, we use local dissipators, which provide a good approximation at short times and for weak couplings in the network compared to the dissipation rates [54–56]. In practice, local dissipators function well up until the ultrastrong coupling regime where the coupling strengths become comparable to the cavity eigenfrequencies [8]. That is, the following results are applicable to systems where the cavity eigenfrequencies dominate over couplings and dissipation rates. In other contexts, global dissipators, which are expressed in terms of the eigenoperators of the coupled system, may be appropriate, for instance, to ensure thermodynamic consistency [57]. We also note that the theory of photon counting statistics that we develop here can readily be extended to global dissipators as well as to multiple baths coupled to the same cavity.

## 2.1 Phase-space description

The steady-state solution of the quantum master equation (3) with the Hamiltonian in Eq. (1) is a Gaussian state [58]. In addition, any Gaussian state remains Gaussian as it evolves according to the quantum dynamics of Eq. (3) [58]. For this reason, we consider the network to be in a Gaussian state at all times. Moreover, Gaussian states are fully characterized by their first two moments, which we define below, and they therefore allow for a compact description.

To provide a phase-space description of the network, we define the characteristic function

$$\chi(t;\boldsymbol{\alpha}) = \mathrm{tr}\big[\hat{\rho}(t)\exp\big(\hat{\mathbf{a}}^\dagger\mathcal{K}\boldsymbol{\alpha}\big)\big], \tag{5}$$

where $\boldsymbol{\alpha} = \big(\alpha_1,\alpha_1^*,\ldots\alpha_N,\alpha_N^*\big)^T$ is a vector of complex variables, and $\mathcal{K}$ is defined above Eq. (2). We note that the Fourier transform of the characteristic function with respect to $\boldsymbol{\alpha}$ is the Wigner function [59–61] [see Eq. (A.10) in App. A.1], which is often used, for instance, for quantum state tomography of bosonic systems [14, 62].

Gaussian states are defined by having a Gaussian characteristic function (and hence their Wigner function is also Gaussian). These states can be written as

$$\chi(t;\boldsymbol{\alpha}) = \exp\left[-\frac{1}{2}\boldsymbol{\alpha}^\dagger\mathcal{K}\Theta(t)\mathcal{K}\boldsymbol{\alpha} + \boldsymbol{d}^\dagger(t)\mathcal{K}\boldsymbol{\alpha}\right], \tag{6}$$

where $\boldsymbol{d}(t) = \langle\hat{\mathbf{a}}\rangle$ is the vector of the first moments, which describe the average displacements of the bosonic modes, and $\Theta$ is the covariance matrix. The covariance matrix is Hermitian with matrix elements that are given by the variances and covariances of the mode operators as

$$[\Theta(t)]_{jk} = \frac{1}{2}\langle[\hat{\mathbf{a}}^\dagger]_j[\hat{\mathbf{a}}]_k + [\hat{\mathbf{a}}]_k[\hat{\mathbf{a}}^\dagger]_j\rangle - \langle[\hat{\mathbf{a}}^\dagger]_j\rangle\langle[\hat{\mathbf{a}}]_k\rangle . \tag{7}$$

We recall that $\hat{\boldsymbol{a}}$ contains the operators $\hat{a}_j$ and $\hat{a}_j^\dagger$ that annihilate and create photons in cavity $j$. For two cavities, for example, the covariance matrix can be concisely expressed using the deviations from the mean, $\delta\hat{a}_j = \hat{a}_j - \langle\hat{a}_j\rangle$, as

$$\Theta(t) = \begin{pmatrix} \langle\delta\hat{a}_1^\dagger\delta\hat{a}_1\rangle + \frac{1}{2} & \langle(\delta\hat{a}_1^\dagger)^2\rangle & \langle\delta\hat{a}_1^\dagger\delta\hat{a}_2\rangle & \langle\delta\hat{a}_1^\dagger\delta\hat{a}_2^\dagger\rangle \\ \langle\delta\hat{a}_1^2\rangle & \langle\delta\hat{a}_1^\dagger\delta\hat{a}_1\rangle + \frac{1}{2} & \langle\delta\hat{a}_1\delta\hat{a}_2\rangle & \langle\delta\hat{a}_1\delta\hat{a}_2^\dagger\rangle \\ \langle\delta\hat{a}_2^\dagger\delta\hat{a}_1\rangle & \langle\delta\hat{a}_2^\dagger\delta\hat{a}_1^\dagger\rangle & \langle\delta\hat{a}_2^\dagger\delta\hat{a}_2\rangle + \frac{1}{2} & \langle(\delta\hat{a}_2^\dagger)^2\rangle \\ \langle\delta\hat{a}_2\delta\hat{a}_1\rangle & \langle\delta\hat{a}_2\delta\hat{a}_1^\dagger\rangle & \langle\delta\hat{a}_2^2\rangle & \langle\delta\hat{a}_2^\dagger\delta\hat{a}_2\rangle + \frac{1}{2} \end{pmatrix}. \tag{8}$$

We stress that the displacement vector $\boldsymbol{d}(t)$ and the covariance matrix $\Theta(t)$ fully describe the state of the bosonic network, both in and out of the equilibrium.

The time evolution of a Gaussian state (6) can be found from the quantum master equation (3), which we discuss in the following section and in App. A.1. Specifically, the vector $\boldsymbol{d}$ of first moments and the covariance matrix $\Theta$ obey the dynamical equations

$$\frac{\mathrm{d}}{\mathrm{d}t}\boldsymbol{d}(t) = \mathcal{A}\boldsymbol{d}(t) + \boldsymbol{f}, \tag{9}$$

and

$$\frac{\mathrm{d}}{\mathrm{d}t}\Theta(t) = \mathcal{A}\Theta(t) + \Theta(t)\mathcal{A}^\dagger + \mathcal{B}. \tag{10}$$

Here, we have introduced the matrix

$$\mathcal{A} = -i\mathcal{K}\mathcal{H} - \gamma/2, \tag{11}$$

which accounts both for the unitary and the dissipative dynamics. We have also defined the diagonal matrices

$$\gamma = \bigoplus_{j=1}^N \gamma_j I_2 = \mathrm{diag}(\gamma_1, \gamma_1, \ldots, \gamma_N, \gamma_N), \tag{12}$$

and

$$\mathcal{B} = \bigoplus_{j=1}^N \gamma_i(\bar{n}_i + 1/2)I_2, \tag{13}$$

which contain the dissipation rates and the temperatures via the Bose-Einstein distributions. Both of these matrices have the block structure of the covariance matrix $\Theta$. We focus on situations, where all eigenvalues of the dynamical matrix $\mathcal{A}$ have non-positive real parts, which ensures the existence of a stable stationary state for the network. This limits the possible values of the parameters $r_j$ and $\lambda_{jk}$ corresponding to single-mode and two-mode squeezing interactions, respectively. We note that Eq. (10) is a Lyapunov equation for the covariance matrix [45, 63].

## 3 Photon counting statistics

In addition to its rich internal dynamics, the bosonic network exchanges photons with its environments. The exchange of photons can be considered as a series of measurements on the network, which interrupt its unitary dynamics. Similarly, the flow of heat through the network between different environments can be understood by the transfer of photons. In both cases, the emission and absorption processes are random, but their statistical distributions contain information about the network as shown in Fig. 2. We note that we are interested in the emission and absorption of photons by the cavities, and we do not need to describe the electromagnetic field outside the cavities. By contrast, in quantum optics, it is common to use

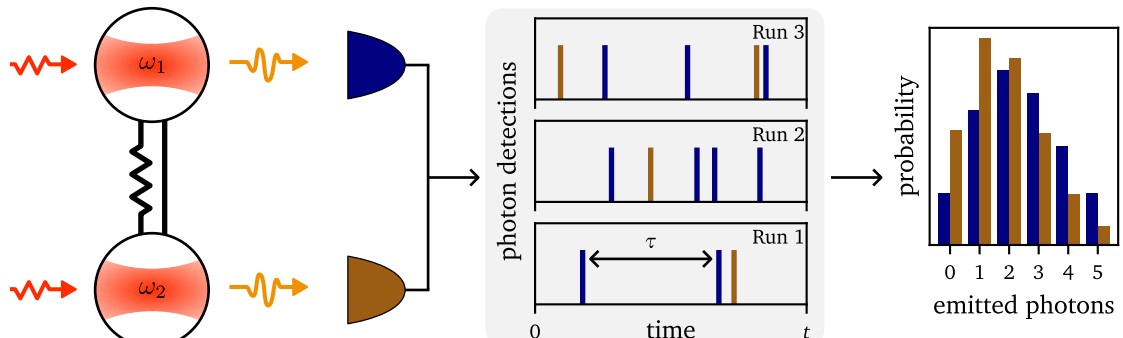

Figure 2: Photon counting statistics. During the time span $[0, t]$ a random number of photons is emitted, depicted by vertical colored lines. The photon counting statistics concerns the distribution of transferred photons and their temporal correlations.

the term "photon counting" to describe how photon-detectors absorb photons from a specific electromagnetic field, and it may not be necessary to describe the source of the field.

The photon counting statistics is encoded in the probability $P(t; \vec{n}, \vec{m})$ for the bosonic network to have emitted $n_j$ photons from cavity $j = 1, \ldots, N$, while having absorbed $m_j$ photons during the time span $[0, t]$. Here, we employ a vector notation so that $\vec{n} = (n_1, \ldots, n_N)$ and $\vec{m} = (m_1, \ldots, m_N)$ contain the number of emitted and absorbed photons in each cavity. We consider the transfer of photons between the cavities and their environments, but not the photons that are exchanged with the coherent fields. The probability distribution can be characterized by its moment-generating function $M(t; \vec{s}, \vec{u})$, which is defined as

$$M(t; \vec{s}, \vec{u}) = \sum_{j=1}^{N} \sum_{n_j=0}^{\infty} \sum_{m_j=0}^{\infty} P(t; \vec{n}, \vec{m}) \exp(\vec{n} \cdot \vec{s} + \vec{m} \cdot \vec{u}), \tag{14}$$

where the vectors of the counting fields, $\vec{s} = (s_1, \ldots, s_N)$ and $\vec{u} = (u_1, \ldots, u_N)$, are conjugate to the number $\vec{n}$ and $\vec{m}$ of emitted and absorbed photons, respectively [64]. For our purposes, it is convenient also to introduce the cumulant generating function, defined as

$$K(t; \vec{s}, \vec{u}) = \ln[M(t; \vec{s}, \vec{u})]. \tag{15}$$

All cumulants of the photon counting statistics can then be obtained by differentiation with respect to the components of $\vec{s}$ or $\vec{u}$ evaluated at zero. For example, the first two cumulants, the averages and the covariances, for the number of emitted photons are

$$\mathrm{E}(n_j)(t) = \left.\frac{\partial M(t; \vec{s}, \vec{u})}{\partial s_j}\right|_{\vec{s}=\vec{u}=0}, \quad \mathrm{Cov}(n_j, n_k)(t) = \left.\frac{\partial^2 K(t; \vec{s}, \vec{u})}{\partial s_j \partial s_k}\right|_{\vec{s}=\vec{u}=0}, \tag{16}$$

with similar expressions for photon absorptions. For a multivariate Gaussian distribution, the first and the second cumulants are the only non-zero cumulants. However, although the state of the network is Gaussian, the photon counting statistics is typically non-Gaussian.

The photon counting statistics can be recovered by inverting Eq. (14) as

$$P(t; \vec{n}, \vec{m}) = \prod_{j=1}^{N} \int_{-i\pi}^{i\pi} \int_{-i\pi}^{i\pi} \frac{\mathrm{d}s_j}{2\pi i} \frac{\mathrm{d}u_j}{2\pi i} \exp[K(t; \vec{s}, \vec{u}) - \vec{n} \cdot \vec{s} - \vec{m} \cdot \vec{u}], \tag{17}$$

where we integrate over the counting fields along the imaginary axis. We note that it is also common to consider counting fields of the form $\vec{s} = i\vec{s}', \vec{u} = i\vec{u}'$ with $\vec{s}'$ and $\vec{u}'$ being real [65].

The photon counting statistics can be obtained from the quantum master equation (3) formulated in terms of elements of the density matrix $\hat{\rho}(t; \vec{n}, \vec{m})$, which has been resolved with respect to the number of photons that have been emitted and absorbed [65, 66]. These photon-resolved density matrices decompose the network's density matrix as $\hat{\rho}(t) = \sum_{j=1}^{N} \sum_{n_j} \sum_{m_j} \hat{\rho}(t; \vec{n}, \vec{m})$. By Fourier transforming this number-resolved master equation with respect to $\vec{n}$ and $\vec{m}$, with the counting fields $\vec{s}$ and $\vec{u}$ as respective conjugate variables, we arrive at the generalized master equation

$$\frac{\mathrm{d}}{\mathrm{d}t} \hat{\rho}(t; \vec{s}, \vec{u}) = \mathcal{L}(\vec{s}, \vec{u}) \hat{\rho}(t; \vec{s}, \vec{u}) \tag{18}$$

$$= \mathcal{L} \hat{\rho}(t; \vec{s}, \vec{u}) + \sum_{j=1}^{N} \gamma_j \Big[ (\bar{n}_j + 1)(e^{s_j} - 1) \hat{a}_j \hat{\rho}(t; \vec{s}, \vec{u}) \hat{a}_j^\dagger + \bar{n}_j (e^{u_j} - 1) \hat{a}_j^\dagger \hat{\rho}(t; \vec{s}, \vec{u}) \hat{a}_j \Big],$$

which now includes the counting fields. The solution of the generalized master equation yields the cumulant generating function of the photon emissions and absorptions as

$$K(t; \vec{s}, \vec{u}) = \ln \operatorname{tr}[\hat{\rho}(t; \vec{s}, \vec{u})], \tag{19}$$

where the introduction of the counting fields breaks the usual normalization of the density matrix. If we set the counting fields to zero, $\vec{s} = \vec{u} = 0$, we recover the original quantum master equation in Eq. (3). Above, we have assumed that all transferred photons are counted. To account for a finite detector efficiency of photons emitted from cavity $j$, one can replace $e^{s_j} - 1$ by $\eta_j(e^{s_j} - 1)$ with $\eta_j < 1$ in Eq. (18). Moreover, if one is interested in the factorial moments and cumulants of the photon counting statistics, rather than the ordinary ones, one can replace the counting factors $e^{s_j} - 1$ and $e^{u_j} - 1$ in Eq. (18) by $s_j$ and $u_j$, respectively [67–69].

## 3.1 Phase-space representation

We now transform the generalized quantum master equation (18) to the phase-space representation. The central idea is that the counting fields do not alter the underlying mathematical structure of the quantum master equation, and generalized Gaussian states exist that enable a significant simplification of Eq. (18). Below, we focus on the main results and defer the lengthy derivation and a discussion of alternative phase-space representations to App. A.1.

To start with, we denote the characteristic function (5) of $\hat{\rho}(t; \vec{s}, \vec{u})$ by $\chi(t; \vec{s}, \vec{u}; \boldsymbol{\alpha})$. The cumulant generating function of Eq. (19) can then be obtained as

$$K(t; \vec{s}, \vec{u}) = \ln \chi(t; \vec{s}, \vec{u}; \boldsymbol{\alpha} = 0). \tag{20}$$

Next, by mapping bosonic operators to differential operators in $\boldsymbol{\alpha}$, the generalized quantum master equation (18) transforms to a partial differential equation for the characteristic function,

$$\frac{\mathrm{d}}{\mathrm{d}t} \chi = \left\{ i \boldsymbol{\alpha}^T \mathcal{H}^T \mathcal{K} \partial_{\boldsymbol{\alpha}} - \boldsymbol{f}^\dagger \boldsymbol{\alpha} + \sum_{j=1}^{N} \gamma_j \Big[ A_j + B_j \Big( \alpha_j \partial_{\alpha_j} + \alpha_j^* \partial_{\alpha_j^*} \Big) + C_j \partial_{\alpha_j} \partial_{\alpha_j^*} \Big] \right\} \chi, \tag{21}$$

where we have introduced a vector of derivatives

$$\partial_{\boldsymbol{\alpha}} = \Big( \partial_{\alpha_1}, \partial_{\alpha_1^*}, \dots \Big)^T, \tag{22}$$

and we have defined three functions reading

$$A_j = -\Big[ (2\bar{n}_j + 1) |\alpha_j|^2 + (\bar{n}_j + 1)(|\alpha_j|^2/2 + 1)(e^{s_j} - 1) + \bar{n}_j (|\alpha_j|^2/2 - 1)(e^{u_j} - 1) \Big]/2,$$
$$B_j = -\Big[ 1 + (\bar{n}_j + 1)(e^{s_j} - 1) - \bar{n}_j (e^{u_j} - 1) \Big]/2, \tag{23}$$
$$C_j = -\Big[ (\bar{n}_j + 1)(e^{s_j} - 1) + \bar{n}_j (e^{u_j} - 1) \Big].$$

With vanishing counting fields, these expressions reduce to $A_j = -(\bar{n}_j + 1/2)|\alpha_j|^2$, $B_j = -1/2$, and $C_j = 0$, and we recover a well-known phase-space representation of the quantum master equation [61]. As such, the counting fields give rise to new terms in the phase-space representation. Moreover, at zero temperature, where $\bar{n}_j = 0$, no photons are absorbed from the environments, and all terms containing the counting fields for absorption $\vec{u}$ vanish.

## 3.2 Riccati equation

In order to solve Eq. (21) for the characteristic function, we proceed with the Ansatz

$$\chi_{\text{ans}}(t; \vec{s}, \vec{u}; \boldsymbol{\alpha}) = \exp\left[-\boldsymbol{\alpha}^\dagger \mathcal{K} \Theta(t; \vec{s}, \vec{u}) \mathcal{K} \boldsymbol{\alpha}/2 + \boldsymbol{d}^\dagger(t; \vec{s}, \vec{u}) \mathcal{K} \boldsymbol{\alpha} + K(t; \vec{s}, \vec{u})\right], \tag{24}$$

which generalizes Eq. (6) by including the counting fields. The cumulant generating function can be obtained as $K(t; \vec{s}, \vec{u}) = \ln \chi_{\text{ans}}(t; \vec{s}, \vec{u}; 0)$ by setting $\boldsymbol{\alpha} = 0$ in the Ansatz. We assume that the network at $t = 0$, when the counting of photons begins, is in a Gaussian state. Therefore, the initial values are $K(0; \vec{s}, \vec{u}) = 0$ together with $\Theta(0; \vec{s}, \vec{u}) = \Theta_0$ and $\boldsymbol{d}(0; \vec{s}, \vec{u}) = \boldsymbol{d}_0$. In Sec. 4, we obtain $\Theta_0$ and $\boldsymbol{d}_0$ from the steady-state solution of the original quantum master equation (3).

We find a system of coupled first-order differential equations for $\Theta, \boldsymbol{d}$, and $K$ by inserting the Ansatz into Eq. (21) and collecting terms to same order in $\boldsymbol{\alpha}$. For the sake of brevity, we omit the arguments $(t; \vec{s}, \vec{u})$ in $\Theta, \boldsymbol{d}$, and $K$, and the system of equations then reads

$$\frac{\mathrm{d}}{\mathrm{d}t}\Theta = \Theta(\Gamma_s + \Gamma_u)\Theta + \mathcal{W}\Theta + \Theta\mathcal{V} + \mathcal{Q}, \tag{25a}$$

$$\frac{\mathrm{d}}{\mathrm{d}t}\boldsymbol{d} = [\mathcal{W} + \Theta(\Gamma_s + \Gamma_u)]\boldsymbol{d} + \boldsymbol{f}, \tag{25b}$$

$$\frac{\mathrm{d}}{\mathrm{d}t}K = \mathrm{tr}[\Gamma_s(\Theta - I_{2N}/2) + \Gamma_u(\Theta + I_{2N}/2)]/2 + \boldsymbol{d}^\dagger(\Gamma_s + \Gamma_u)\boldsymbol{d}/2, \tag{25c}$$

where we have defined the diagonal matrices containing the counting fields

$$\Gamma_s = \bigoplus_{j=1}^{N} \gamma_j(\bar{n}_j + 1)(e^{s_j} - 1)I_2, \quad \Gamma_u = \bigoplus_{j=1}^{N} \gamma_j \bar{n}_j(e^{u_j} - 1)I_2, \tag{26}$$

as well as the matrices

$$\mathcal{W} = \mathcal{A} - (\Gamma_s - \Gamma_u)/2, \quad \mathcal{V} = \mathcal{A}^\dagger - (\Gamma_s - \Gamma_u)/2, \quad \mathcal{Q} = \mathcal{B} + (\Gamma_s + \Gamma_u)/4, \tag{27}$$

where the expressions for $\mathcal{A}$ and $\mathcal{B}$ can be found in Eqs. (11,13).

This set of coupled equations constitutes the main theoretical result of our work, and it allows us to fully characterize the photon counting statistics of a bosonic network on all relevant time scales. Specifically, the full time-dependent photon counting statistics can be obtained from the solution for $K(t; \vec{s}, \vec{u})$. We can make several observations about these equations:

1. The counting fields in $\Gamma_s$ and $\Gamma_u$ produce a term that is of second order in the covariance matrix $\Theta$. The dynamical equation for $\Theta$ is known as a matrix Riccati equation, and it has been investigated extensively in optimal control theory [70–72].

2. The dynamical equations have a cascading structure, where one can first solve for $\Theta(t; \vec{s}, \vec{u})$, then use this solution to solve for $\boldsymbol{d}(t; \vec{s}, \vec{u})$, and finally use both solutions to find $K(t; \vec{s}, \vec{u})$.

3. If there is no coherent drive, $\boldsymbol{f} = 0$, we have $\boldsymbol{d}_0 = 0$ and $\boldsymbol{d}(t; \vec{s}, \vec{u}) = 0$ at all times. The cumulant generating function can then be obtained from the covariance matrix only.

4. Without photon counting, we have $\Gamma_s = \Gamma_u = 0$ and $K(t; \vec{s}, \vec{u}) = 0$, and we recover Eqs. (9,10) for $\Theta$ and $\boldsymbol{d}$, which are uncoupled.

In addition to these observations, we note that the coupled equations allow for a systematic expansion for the time evolution of the cumulants of the photon counting statistics. For photon emissions, the expansion to second order in the counting fields is discussed in App. A.5. Generally, we can expand $\Gamma_s$ in the emission counting fields as

$$\Gamma_s = \sum_{j=1}^{N} \sum_{n_j=1}^{\infty} \frac{s_j^{n_j}}{n_j!} \Gamma_j \, , \tag{28}$$

where the matrices $\Gamma_j = \bigoplus_{k=1}^{N} \delta_{jk} \gamma_k (\bar{n}_k + 1) I_2$ do not depend on the counting fields, and $\delta_{jk}$ is the Kronecker delta. A similar expansion in the absorption counting fields can be made for $\Gamma_u$.

## 3.3 Short-time statistics

Using the framework developed above, we can investigate the correlations of the photon counting statistics encoded in the distribution of waiting times between photon emissions and the $g^{(2)}$-correlation function. As we will see, they are both directly related to the set of equations in Eq. (25). Thus, we consider the probability density that the emission of a photon from mode $j$ at the time $t = 0$ is followed by another emission from mode $k$ at the later time $t = \tau$. If we impose that no other photons are emitted from mode $k$ during the time span $[0, \tau]$, we are describing the waiting time distribution $W_{jk}(\tau)$ of the photon emissions. Without this condition, the corresponding quantity is the second-order coherence function $g^{(2)}_{jk}(\tau)$.

In both cases, it will be useful to define the correlation function

$$c_{jk}(\tau; \vec{s}) = \text{tr}\left( \mathcal{J}_k e^{\mathcal{L}(\vec{s},0)\tau} \mathcal{J}_j \hat{\rho}_0 \right), \tag{29}$$

in terms of the superoperator for photon emission

$$\mathcal{J}_j \hat{\rho} = \gamma_j (\bar{n}_j + 1) \hat{a}_j \hat{\rho} \hat{a}_j^\dagger \, . \tag{30}$$

Here, the steady-state density matrix is the solution to the eigenproblem $\mathcal{L} \hat{\rho}_0$ with $\mathcal{L} = \mathcal{L}(0,0)$, and $\mathcal{L}(\vec{s}, \vec{u})$ is the Liouvillian of the generalized master equation (18). The condition that no photons are emitted from mode $k$ can be imposed by setting $s_k \to -\infty$ or, equivalently, $e^{s_k} \to 0$. To see this, one may consider the definition of the moment generating function in Eq. (14) and set all counting fields to zero except for $s_k$. We then see that $M(\tau; s_k \to -\infty) = P(\tau; n_k = 0)$ indeed is the probability that no photons are emitted from mode $k$ during the time span $[0, \tau]$. Here and below, we only write the counting fields that are non-zero and leave out those that are set to zero. The waiting time distribution then becomes [61, 65]

$$W_{jk}(\tau) = c_{jk}(\tau; s_k \to -\infty)/J_j \, , \tag{31}$$

while the normalized second-order coherence function reads

$$g^{(2)}_{jk}(\tau) = c_{jk}(\tau)/J_j J_k \, , \tag{32}$$

where we have defined the average photon currents in the steady state, $J_{j,k} = \text{tr}\left( \mathcal{J}_{j,k} \hat{\rho}_0 \right)$. We see that the waiting time distribution indeed is a probability density, since it integrates to one, $\int_0^\infty d\tau \, W_{jk}(\tau) = 1$ [65]. The second-order coherence function, by contrast, is dimensionless. We also see that the two quantities are related at $\tau = 0$, where we have

$$W_{jk}(0) = \gamma_k \langle \hat{a}_j^\dagger \hat{a}_k^\dagger \hat{a}_k \hat{a}_j \rangle / \langle \hat{a}_j^\dagger \hat{a}_j \rangle \, , \tag{33}$$

for the waiting time distribution, while the second-order coherence function becomes

$$g_{jk}^{(2)}(0) = \langle \hat{a}_j^\dagger \hat{a}_k^\dagger \hat{a}_k \hat{a}_j \rangle / (\langle \hat{a}_j^\dagger \hat{a}_j \rangle \langle \hat{a}_k^\dagger \hat{a}_k \rangle), \tag{34}$$

and we can conclude that

$$W_{jk}(0) = J_k g_{jk}^{(2)}(0). \tag{35}$$

We use our phase-space approach to evaluate the correlator in Eq. (29). Specifically, we use that the steady-state density matrix of the bosonic network and its characteristic function $\chi_0$ are both Gaussian, although the state right after a photon emission is not. The time-evolution generated by $\exp[\mathcal{L}(\vec{s}, 0)\tau]$ corresponds to the time-evolution generated by Eq. (21) for any characteristic function. Therefore, we can define a phase-space time evolution operator by

$$\mathcal{T}(\tau; \vec{s})\chi(t = 0; \boldsymbol{\alpha}) = \chi(\tau; \vec{s}; \boldsymbol{\alpha}), \tag{36}$$

and identify a representation for the correlation function

$$c_{jk}(\tau; \vec{s}) = D_k \mathcal{T}(\tau; \vec{s}) D_j \chi_0(\boldsymbol{\alpha})\big|_{\boldsymbol{\alpha}=0}, \tag{37}$$

where $D_{j,k}$ are the differential operators

$$D_j = -\gamma_j(\bar{n}_j + 1)\Big(1 + |\alpha_j|^2/2 + \alpha_j \partial_{\alpha_j} + \alpha_j^* \partial_{\alpha_j^*} + 2\partial_{\alpha_j}\partial_{\alpha_j^*}\Big)/2, \tag{38}$$

corresponding to the emission superoperators $\mathcal{J}_{j,k}$.

Next, one has to evolve the quantity $D_j\chi_0(\boldsymbol{\alpha})$ forward in time. Assuming a Gaussian characteristic function, we have $D_j\chi_0(\boldsymbol{\alpha}) = F(\boldsymbol{\alpha})\chi_0(\boldsymbol{\alpha})$, where $F(\boldsymbol{\alpha})$ is a polynomial function of $\boldsymbol{\alpha}$ that also depends on the coherence matrix $\Theta_0$ and the vector of the first moments $\boldsymbol{d}_0$. The polynomial $F$ describes that the state right after an emission is not Gaussian. We solve the time evolution of this quantity by assuming that it can be written as $f(\tau; \vec{s}; \boldsymbol{\alpha})\chi_{\text{ans}}(\tau; \vec{s}; \boldsymbol{\alpha})$. Here, $\chi_{\text{ans}}$ is the Gaussian state Ansatz (24), whose time evolution is governed by Eq. (25), and $f$ is an additional polynomial Ansatz such that $f(0; \vec{s}; \boldsymbol{\alpha}) = F(\boldsymbol{\alpha})$. Hence, this Ansatz gives an additional system of equations. The details of this derivation are given in App. A.3.

For the correlation function, we find

$$c_{jk}(\tau; \vec{s}) = e^{K(\tau; \vec{s})}\Big\{\big[J_j + z(\tau; \vec{s})\big]J_k(\tau; \vec{s}) + \text{tr}[\Gamma_k X(\tau; \vec{s})]/2 + \boldsymbol{d}^\dagger(\tau; \vec{s})\Gamma_k \boldsymbol{y}(\tau; \vec{s})\Big\}, \tag{39}$$

where the conditional mean photon current from mode $k$ reads

$$J_k(t; \vec{s}) = \text{tr}[\Gamma_k \Theta_N(t; \vec{s})]/2 + \boldsymbol{d}^\dagger(t; \vec{s})\Gamma_k \boldsymbol{d}(t; \vec{s})/2, \tag{40}$$

and $\Theta_N = \Theta - I_{2N}/2$ is the normal-ordered covariance matrix.

The quantities $\Theta, \boldsymbol{d}$, and $K$ follow Eq. (25), whereas $X, \boldsymbol{y}$, and $z$ obey the equations

$$\frac{\mathrm{d}}{\mathrm{d}t}X(t; \vec{s}) = [\mathcal{A} + \Theta_N(t; \vec{s})\Gamma_s]X(t; \vec{s}) + X(t; \vec{s})[\mathcal{A} + \Theta_N(t; \vec{s})\Gamma_s]^\dagger, \tag{41a}$$

$$\frac{\mathrm{d}}{\mathrm{d}t}\boldsymbol{y}(t; \vec{s}) = [\mathcal{A} + \Theta_N(t; \vec{s})\Gamma_s]\boldsymbol{y}(t; \vec{s}) + X(t; \vec{s})\Gamma_s \boldsymbol{d}(t; \vec{s}), \tag{41b}$$

$$\frac{\mathrm{d}}{\mathrm{d}t}z(t; \vec{s}) = \text{tr}[\Gamma_s X(t; \vec{s})]/2 + \boldsymbol{d}^\dagger(t; \vec{s})\Gamma_s \boldsymbol{y}(t; \vec{s}), \tag{41c}$$

with the initial conditions $X(0; \vec{s}) = \Theta_{N,0}\Gamma_j\Theta_{N,0}$, $\boldsymbol{y}(0; \vec{s}) = \Theta_{N,0}\Gamma_j\boldsymbol{d}_0$, and $z(0; \vec{s}) = 0$, where $\Theta_{N,0} = \Theta_0 - I_{2N}/2$ is the normal-ordered steady-state covariance matrix. We note that the matrix $\Gamma_s$ depends on the time evolution operator $\mathcal{T}(\tau; \vec{s})$ through Eq. (25).

The correlator $c_{jk}(t; \vec{s})$ can now be evaluated by solving Eqs. (25) and (41). For the waiting time distributions, these sets of equations can be solved analytically or numerically. Moreover,

the short- and long-time behavior can be evaluated systematically as discussed in App. A.3. The equations for the waiting time distribution simplify, if one is interested in emissions from one specific cavity only and hence sets $j = k$. In that case, we have $X(t; \vec{s}) = -\frac{\mathrm{d}}{\mathrm{d}t}\Theta_N(t; \vec{s})$ and $\boldsymbol{y}(t; \vec{s}) = -\frac{\mathrm{d}}{\mathrm{d}t}\boldsymbol{d}(t; \vec{s})$, since $\Gamma_s \to -\Gamma_j$, when evaluating the time evolution operator in $c_{jj}(\tau; s_j \to -\infty)$. The integration of the variable $z$ then relates it to the deviation of the mean photon current caused by the condition of no photon emissions as $z(t; \vec{s}) = J_j(t; s_j \to -\infty) - J_j$.

These results can be extended to joint waiting time distributions involving several photon emissions. For example, one can consider the probability density of first observing a waiting time of duration $\tau_1$ between a photon emission from mode $j$ and a photon emission from mode $k$, followed by a subsequent waiting time of duration $\tau_2$ until a third photon emission from mode $m$. Each waiting time would then correspond to a set of differential equations.

For the second-order coherence functions, we set $\vec{s} = 0$ and use $\Theta(t) = \Theta_0$ and $\boldsymbol{d}(t) = \boldsymbol{d}_0$. We then find $z(t) = K(t) = 0$ as well as $X(t) = e^{\mathcal{A}t}X(0)e^{\mathcal{A}^{\dagger}t}$ and $\boldsymbol{y}(t) = e^{\mathcal{A}t}\boldsymbol{y}(0)$. The second-order coherence functions then become

$$g_{jk}^{(2)}(\tau) = 1 + \left[\mathrm{tr}\left(\Gamma_k e^{\mathcal{A}\tau}\Theta_{N,0}\Gamma_j\Theta_{N,0}e^{\mathcal{A}^{\dagger}\tau}\right)/2 + \boldsymbol{d}_0^{\dagger}\Gamma_k e^{\mathcal{A}\tau}\Theta_{N,0}\Gamma_j\boldsymbol{d}_0\right]/J_j J_k, \tag{42}$$

which would also follow from the quantum regression theorem [61, 73]. We note that $g_{jk}^{(2)}(\tau) = g_{kj}^{(2)}(-\tau)$ due to the stationarity of the steady-state emission processes.

The second-order coherence functions and the waiting time distributions differ in how they depend on a finite detection efficiency. The finite detector efficiency $\eta_j < 1$ modifies the emission counting fields, which changes $\Gamma_j \to \eta_j \Gamma_j$ in the previous equations. Then, it can be shown that the second-order coherence function $g_{jk}^{(2)}$ is independent of the detector efficiencies $\eta_{j,k}$ whereas the WTD $W_{jk}$ is changed in a nonlinear manner. Therefore, a measurement with imperfect detectors directly gives the second-order coherence function, while the actual waiting time distribution has to be estimated. We discuss this further for the two-cavity system in Sec. 4.

## 3.4  Long-time statistics

In addition to the short-time dynamics encoded in the waiting time distributions and the second-order coherence functions, we may consider the photon counting statistics at long times. Specifically, we now focus on observation times $[0, t]$ that are much longer than any other time scales. We can then neglect the transients of the differential equations (25) and evaluate the photon counting statistics at long times. In this limit, the cumulant generating function becomes linear in time, such that $K(t; \vec{s}, \vec{u}) = t\tilde{K}(\vec{s}, \vec{u})$, where $\tilde{K}$ is the scaled cumulant generating function, which is given by the eigenvalue of the modified Liouvillian $\mathcal{L}(t; \vec{s}, \vec{u})$ with the largest real part [65]. All other eigenvalues have smaller real parts, and their relative contribution to the cumulant generating function decays exponentially with time.

To find the cumulant generating function, we set $\frac{\mathrm{d}}{\mathrm{d}t}\Theta(t; \vec{s}, \vec{u}) = 0$ and $\frac{\mathrm{d}}{\mathrm{d}t}\boldsymbol{d}(t; \vec{s}, \vec{u}) = 0$ in Eq. (25). There are then several approaches to find the covariance matrix $\Theta$ from the algebraic Riccati equation in Eq. (25a). Here, we use a method from Ref. [74], which is similar to completing the square. Since the cumulant generating function admits a unique Taylor expansion around $\vec{s} = \vec{u} = 0$, we can for now treat the matrices $\Gamma_s$ and $\Gamma_u$ as being real valued. We then have $\mathcal{V} = \mathcal{W}^{\dagger}$ in Eq. (25), which simplifies the solution by ensuring that $\Theta(\vec{s}, \vec{u})$ is Hermitian. Then, if we are able to find a matrix $\mathcal{F}$ such that

$$\mathcal{F}\mathcal{F}^{\dagger} = \mathcal{W}(\Gamma_s + \Gamma_u)^{-1}\mathcal{W}^{\dagger} - \mathcal{Q}, \tag{43}$$

the time-independent solution becomes

$$\Theta(\vec{s}, \vec{u}) = \mathcal{F}(\Gamma_s + \Gamma_u)^{-1/2} - \mathcal{W}(\Gamma_s + \Gamma_u)^{-1}, \tag{44}$$

where the square root of a matrix is defined as $\Gamma^{1/2}\Gamma^{1/2} = \Gamma$. There are multiple solutions for $\mathcal{F}$, and we can identify the appropriate one by imposing the condition that $\tilde{K}(0,0) = 0$.

The steady-state solution for the first moments $\boldsymbol{d}$ readily follows by matrix inversion. However, there is a connection to the matrix $\mathcal{F}$, which simplifies the solution. To see this, we note that we can write $\mathcal{W} + \Theta(\Gamma_s + \Gamma_u) = \mathcal{F}(\Gamma_s + \Gamma_u)^{1/2}$ on the right-hand side of the equation for $\boldsymbol{d}$ in Eq. (25b). By using this relation together with the definition of $\mathcal{F}$, we find

$$\boldsymbol{d}^\dagger(\Gamma_s + \Gamma_u)\boldsymbol{d} = \boldsymbol{f}^\dagger\big[\mathcal{W}(\Gamma_s + \Gamma_u)^{-1}\mathcal{W}^\dagger - \mathcal{Q}\big]^{-1}\boldsymbol{f}\,. \tag{45}$$

Having found the covariance matrix and the vector of first moments, we then obtain the scaled cumulant generation function from the last line of Eq. (25). Specifically, we have

$$\begin{aligned}
\tilde{K}(\vec{s},\vec{u}) = {}&\mathrm{tr}[\Gamma_s(\Theta(\vec{s},\vec{u}) - I_{2N}/2) + \Gamma_u(\Theta(\vec{s},\vec{u}) + I_{2N}/2)]/2 \\
&+ \boldsymbol{f}^\dagger\big[\mathcal{W}(\Gamma_s + \Gamma_u)^{-1}\mathcal{W}^\dagger - \mathcal{Q}\big]^{-1}\boldsymbol{f}/2\,,
\end{aligned} \tag{46}$$

where $\Theta(\vec{s},\vec{u})$ is given in Eq. (44). This expression illustrates the complex interplay between the coherent drive ($\boldsymbol{f}$), the unitary and dissipative dynamics ($\mathcal{W}$), and the thermal noise ($\mathcal{Q}$).

The first two cumulants of the photon emission statistics in the long-time limit can be expressed in terms of the short-time statistics. Using an expansion of Eq. (46) in the counting fields, or alternatively Eq. (25) as shown in App. A.5, and the second-order coherence function in Eq. (42), one finds the well-known expressions for the Fano factors [65]

$$\frac{\mathrm{Cov}(n_j, n_k)}{\mathrm{E}(n_j)} = \delta_{jk} + 2J_k \int_0^\infty \mathrm{d}\tau \left[\frac{g_{jk}^{(2)}(\tau) + g_{kj}^{(2)}(\tau)}{2} - 1\right]\,, \tag{47}$$

where $\mathrm{Cov}(n_j, n_j) = \mathrm{Var}(n_j)$ denotes the variance. This relation arises because the short and long-time statistics are both determined by the fluctuations in the steady state.

# 4 Applications

We are now ready to apply our theoretical framework to specific bosonic networks and to discuss how the photon counting statistics are affected by the interactions between the cavities in the network. In addition to our focus on the photon counting statistics, we will also consider the entanglement between the cavities as well as the transport of photons in a network. We first consider a simple network consisting of just two coupled cavities for which we investigate the time-dependent correlations of the emitted photons and the photon emission statistics at long times. We then discuss how the statistics of emitted photons can be related to the entanglement between the cavities. Finally, we consider the transport properties of a bosonic circulator that is based on three coupled cavities. As shown above, we can always start by evaluating the photon counting statistics without a coherent drive and then include it later, if needed. For illustrative purposes, we focus here on small networks with only two or three coupled cavities. However, our formalism applies equally well for larger networks, and the dimension of the involved matrix equations grows only linearly with the number of cavities.

## 4.1 Two coupled cavities

We first focus on two cavities with the same mode frequency and discuss the non-local correlations between the photons emitted from the cavities. These correlations are generated by the

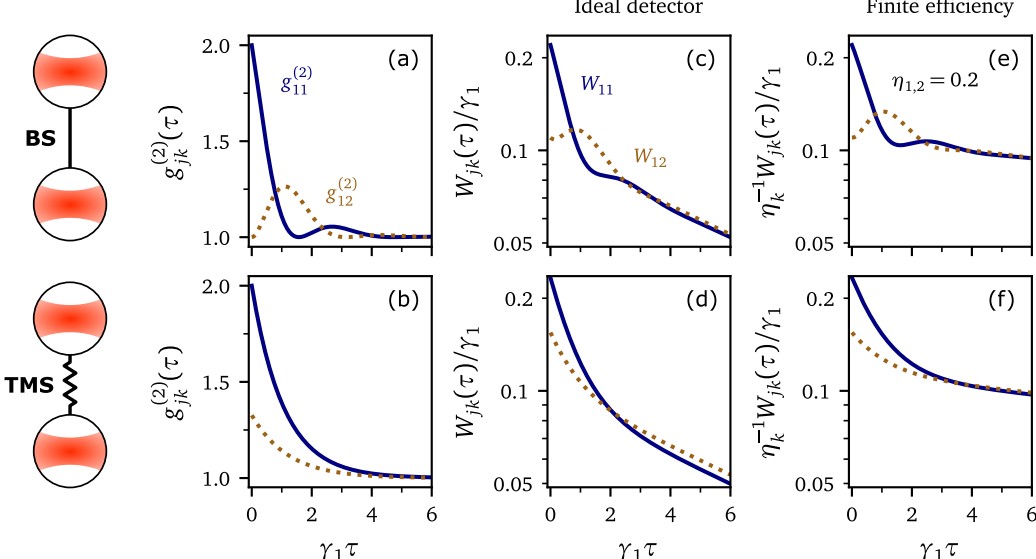

Figure 3: Short-time statistics. (a,b) Second-order coherence functions for BS and TMS interactions. (c–f) Waiting time distributions on a logarithmic scale with (c,d) ideal detectors or (e,f) detectors that only detect 20% of the photons. The vertical axes in panels (e,f) are rescaled by the detection efficiency $\eta_{1,2} = 0.2$. Parameters are $g/\gamma_1 = 1$ (BS coupling), $\lambda/\gamma_1 = 0.05$ (TMS coupling), $\bar{n}_1 = \bar{n}_2 = 0.1$ and $\gamma_1 = \gamma_2$.

beam-splitting (BS) or two-mode squeezing (TMS) interactions given by the Hamiltonians in a frame that is rotating at the cavity eigenfrequencies

$$\hat{H}_{\text{BS}} = g(\hat{a}_1^\dagger \hat{a}_2 + \hat{a}_1 \hat{a}_2^\dagger), \quad \text{and} \quad \hat{H}_{\text{TMS}} = \lambda(\hat{a}_1 \hat{a}_2 + \hat{a}_1^\dagger \hat{a}_2^\dagger), \tag{48}$$

which correspond to the dynamical matrices

$$\mathcal{A}_{\text{BS}} = \begin{pmatrix} -\frac{\gamma_1}{2} & 0 & -ig & 0 \\ 0 & -\frac{\gamma_1}{2} & 0 & ig \\ -ig & 0 & -\frac{\gamma_2}{2} & 0 \\ 0 & ig & 0 & -\frac{\gamma_2}{2} \end{pmatrix}, \quad \text{and} \quad \mathcal{A}_{\text{TMS}} = \begin{pmatrix} -\frac{\gamma_1}{2} & 0 & 0 & -i\lambda \\ 0 & -\frac{\gamma_1}{2} & i\lambda & 0 \\ 0 & -i\lambda & -\frac{\gamma_2}{2} & 0 \\ i\lambda & 0 & 0 & -\frac{\gamma_2}{2} \end{pmatrix}. \tag{49}$$

We can choose the coupling constants to be real without loss of generality, and we may also assume that $2\lambda < \sqrt{\gamma_1 \gamma_2}$, which ensures that all eigenvalues of $A_{\text{TMS}}$ are negative, such that the system has a stable steady state. Furthermore, the cavities can be driven coherently as described by the Hamiltonian

$$\hat{H}_{\text{drive}} = -i \sum_j (f_j^* \hat{a}_j - f_j \hat{a}_j^\dagger), \tag{50}$$

so that the frequency of the coherent drive matches the frequencies of the cavities, but its phase can vary with respect to the those of the couplings.

As discussed in Sec. 3.3, the second-order coherence functions are straightforward to calculate based on Eq. (42). To keep the discussion simple, we set $\bar{n}_1 = \bar{n}_2$ and $\gamma_1 = \gamma_2$, and we then recover the well-known textbook results

$$g_{11,\text{BS}}^{(2)}(\tau) = 1 + \cos^2(g\tau)e^{-\gamma_1 |\tau|},$$
$$g_{12,\text{BS}}^{(2)}(\tau) = 1 + \sin^2(g\tau)e^{-\gamma_1 |\tau|}, \tag{51}$$

for BS interactions, while for the TMS interactions, we find

$$g_{11,\text{TMS}}^{(2)}(\tau) = 1 + \left[\cosh(\lambda\tau) + \frac{\gamma_1}{2\lambda}\frac{\bar{n}_A}{\bar{n}_1 + \bar{n}_A}\sinh(\lambda|\tau|)\right]^2 e^{-\gamma_1|\tau|},$$

$$g_{12,\text{TMS}}^{(2)}(\tau) = 1 + \left[\sinh(\lambda|\tau|) + \frac{\gamma_1}{2\lambda}\frac{\bar{n}_A}{\bar{n}_1 + \bar{n}_A}\cosh(\lambda\tau)\right]^2 e^{-\gamma_1|\tau|},$$

(52)

where $\bar{n}_A = 2(1+\bar{n}_1)\lambda^2/(\gamma_1^2 - 4\lambda^2)$ describes the increased number of photons due to the TMS interaction in the steady state, such that $\langle\hat{a}_1^\dagger\hat{a}_1\rangle = \bar{n}_1 + \bar{n}_A$.

Figure 3 shows the second-order coherence functions, which display clear signatures of photon bunching in all fours cases with $g_{jk}^{(2)}(\tau) > 1$ at all times. Because the cavities have the same frequencies, we have $g_{11}^{(2)}(\tau) = g_{22}^{(2)}(\tau)$ and $g_{12}^{(2)}(\tau) = g_{21}^{(2)}(\tau)$. The coherence functions also have the property that $g^{(2)}(\tau) = g^{(2)}(-\tau)$ according to Eqs. (51) and (52). For the TMS interactions, the cross and autocorrelations are the same for $\lambda = \gamma_1\bar{n}_1$, where we find $g_{12,\text{TMS}}^{(2)}(\tau) = g_{11,\text{TMS}}^{(2)}(\tau)$. In addition, for the TMS interactions, the coherence functions depend on the temperature through $\bar{n}_1$. By contrast, for the BS interactions, the cross and autocorrelations are always out of phase, and they are both independent of the temperature. Moreover, the BS interactions lead to the coherent exchange of photons between the cavities and oscillating coherence functions. Also, the emission of a photon from one cavity affects the state of both cavities. Specifically, right after the emission of a photon, the average number of photons in the cavity is in fact larger than in the steady state. (For a single thermal cavity, there are exactly twice as many photons [39].) However, after a short time span, related to the coupling $g$, the emission from the same cavity becomes less likely. On the other hand, an emission from the other cavity becomes more likely. Similarly, the TMS interactions lead to a large degree of correlations. Specifically, the two cavities contain a correlated number of photons in the steady state. In addition, if one cavity emits a photon, it is increasingly likely that the other cavity will also emit a photon soon after as compared to two independent cavities.

The coherence functions in Fig. 3(a,b) can be contrasted with the waiting time distributions in Fig. 3(c,d). The waiting time distributions are calculated numerically, but can be evaluated analytically in certain limits. The waiting time distributions and the coherence functions are related as short times. By contrast, their long-time behavior is very different. The coherence functions level off to one at long times, since the emissions of photons with a long time span in between are uncorrelated. By contrast, the waiting time distributions decay exponentially, since it is unlikely that two subsequent photon emissions are separated by a long time interval. Thus, the waiting time distributions take the form $W_{jk}(\tau) \simeq e^{-\gamma_{jk}^\infty\tau}$, where the rate $\gamma_{jk}^\infty$ can be related to the scaled cumulant generating function as $\gamma_{jk}^\infty = -\tilde{K}(s_k \to -\infty)$ with the other counting fields set to zero. In Fig. 3, the dissipation rates and the temperatures have been chosen such that the decay rates $\gamma_{jk}^\infty$ are all the same, and we find

$$\gamma_{\text{BS}}^\infty/\gamma_1 = \sqrt{\left(\sqrt{1/4 + g^2/\gamma_1^2} + \sqrt{1/4 + g^2/\gamma_1^2 + \bar{n}_1(\bar{n}_1+1)}\right)^2 - 4g^2/\gamma_1^2} - 1$$

$$\simeq \bar{n}_1 + \frac{g^2}{\gamma_1^2}\frac{2\bar{n}_1^2}{(\bar{n}_1+1)(2\bar{n}_1+1)},$$

(53)

and

$$\gamma_{\text{TMS}}^\infty/\gamma_1 = \sqrt{1/2 + \bar{n}_1 + \bar{n}_1^2 + 2\lambda^2/\gamma_1^2 + \sqrt{(\bar{n}_1+1/2)^2 + 2\lambda^2(1+4\bar{n}_1+2\bar{n}_1^2)/\gamma_1^2 + 4\lambda^4/\gamma_1^4}} - 1$$

$$\simeq \bar{n}_1 + \frac{\lambda^2}{\gamma_1^2}\left(2 - \frac{2\bar{n}_1^2}{(\bar{n}_1+1)(2\bar{n}_1+1)}\right).$$

(54)

The approximations above apply in the weak-coupling regime for both types of interactions. With no coupling between the cavities, we recover the result for a single thermal cavity $\gamma^\infty = \gamma_1 \bar{n}_1$ [39]. Moreover, at high temperatures, $\bar{n}_1 \gg 1$, the decay rate behaves similarly for both types of interactions as $\gamma_{\mathrm{BS}}^\infty/\gamma_1 \simeq \bar{n}_1 + g^2/\gamma_1^2$ and $\gamma_{\mathrm{TMS}}^\infty/\gamma_1 \simeq \bar{n}_1 + \lambda^2/\gamma_1^2$.

If some photons are not detected, the waiting time distribution becomes flatter as seen in Figs. 3(e,f). The distribution becomes smaller at short waiting times but in turn extends to longer waiting times. It can be shown from the definition in Eq. (31) that, at zero waiting time, the measured distribution is related to the actual one as $W_{jk}(0) = \eta_k W_{jk}^{\eta_{1,2}=1}(0)$. Similarly, the decay rate of the distribution with the detection efficiency $\eta$ at high temperatures, corresponding to the weak-coupling limit of Eqs. (53,54), is given by $\gamma_{\mathrm{BS}}^\infty/\gamma_1 \simeq \sqrt{\eta}\bar{n}_1 + (\sqrt{\eta}-1)/2 + g^2/\gamma_1^2$ and $\gamma_{\mathrm{TMS}}^\infty/\gamma_1 \simeq \sqrt{\eta}\bar{n}_1 + (\sqrt{\eta}-1)/2 + \lambda^2/\gamma_1^2$. Despite a low detection efficiency, the waiting time distributions and the second-order coherence functions share several characteristics.

Having considered the photon counting statistics at short times, we now turn to the limit of long observation times. The photon emission statistics of two cavities is most naturally expressed in terms of the normal-ordered covariance matrix $\Theta_N = \Theta - I_4/2$. Setting the absorption counting fields $\vec{u}$ to zero, its dynamical equation derived from Eq. (25a) reads

$$\frac{\mathrm{d}}{\mathrm{d}t}\Theta_N(t;\vec{s}) = \Theta_N(t;\vec{s})\Gamma_s\Theta_N(t;\vec{s}) + \mathcal{A}\Theta_N(t;\vec{s}) + \Theta_N(t;\vec{s})\mathcal{A}^\dagger + \mathcal{B}_N, \tag{55}$$

where we have introduced $\mathcal{B}_N = \mathcal{B} - \gamma/2 + i(\mathcal{HK} - \mathcal{KH})/2$. The matrices $\mathcal{A}$ and $\mathcal{B}$ are those that enter the dynamical equation for $\Theta$ without the counting fields, and they are defined in Eqs. (11)–(13). We can then use the method from Sec. 3.4 to solve for $\Theta_N$ in the cumulant generating function. The calculation simplifies because the unknown matrix $\mathcal{F}$ that enters in $\Theta_N$ inherits the structure of $\mathcal{A}$ from the condition that $\mathcal{F}\mathcal{F}^\dagger = \mathcal{A}\Gamma_s^{-1}\mathcal{A}^\dagger - \mathcal{B}_N$.

After some algebra, which is described in App. A.4, we find the cumulant generating function at long times. For BS interactions without a coherent drive, it reads

$$\tilde{K}_{\mathrm{BS}}(\vec{s}) = \lim_{t\to\infty}\frac{K_{\mathrm{BS}}(t;\vec{s})}{t} = \frac{\gamma_1+\gamma_2}{2} - \sqrt{\frac{\xi_1}{2} + \frac{\xi_2}{2} - \frac{(\gamma_1+\gamma_2)^2}{\gamma_1\gamma_2}g^2 + \sqrt{\xi_1\xi_2 + 4\gamma_1\gamma_2 g^2 F_g}}, \tag{56}$$

where we have defined

$$\xi_i = \frac{1}{2}\gamma_i^2[1 + \frac{4}{\gamma_1\gamma_2}g^2 - 4\bar{n}_i(\bar{n}_i+1)(e^{s_i}-1)], \tag{57}$$

and

$$F_g = (\bar{n}_1 - \bar{n}_2)[(\bar{n}_1+1)(e^{s_1}-1) - (\bar{n}_2+1)(e^{s_2}-1)]. \tag{58}$$

Here, we see how the BS interactions generate an intricate structure in the cumulant generating function, which manifests itself in non-zero cross-correlations. Also, setting $g = 0$, we recover the cumulant generating function of two independent cavities, $\tilde{K} = \sum_{i=1}^2 (\gamma_i - \sqrt{2\xi_i})/2$.

For TMS interactions, we similarly find

$$\tilde{K}_{\mathrm{TMS}}(\vec{s}) = \frac{\gamma_1+\gamma_2}{2} - \sqrt{\frac{\zeta_1}{2} + \frac{\zeta_2}{2} + \lambda^2\frac{(\gamma_1+\gamma_2)^2}{\gamma_1\gamma_2} + \sqrt{\zeta_1\zeta_2 - 4\gamma_1\gamma_2\lambda^2 F_\lambda}}, \tag{59}$$

where we defined

$$\zeta_i = \frac{1}{2}\gamma_i^2[1 - \frac{4}{\gamma_1\gamma_2}\lambda^2 - 4\bar{n}_i(\bar{n}_i+1)(e^{s_i}-1)], \tag{60}$$

and

$$F_\lambda = (\bar{n}_1+1)(\bar{n}_2+1)(e^{s_1}-1)(e^{s_2}-1) + (1+\bar{n}_1+\bar{n}_2)[(\bar{n}_1+1)(e^{s_1}-1) + (\bar{n}_2+1)(e^{s_2}-1)]. \tag{61}$$

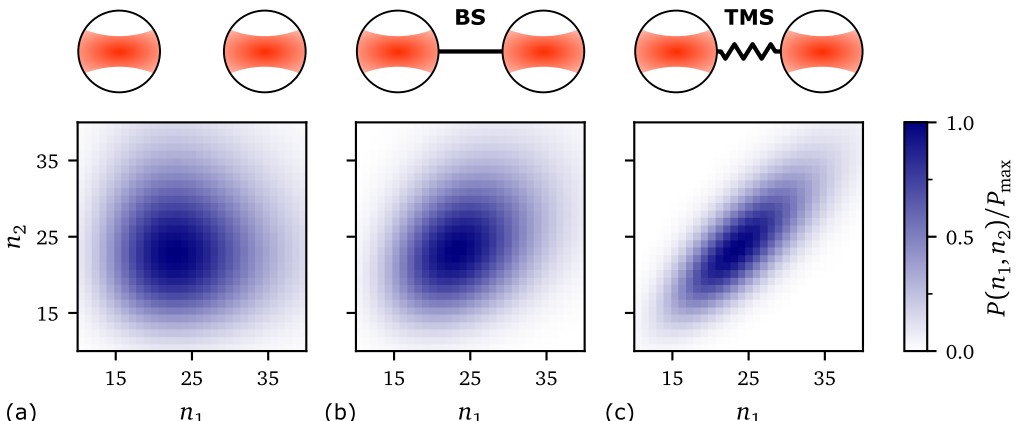

Figure 4: Long-time statistics. Distributions of emitted photons from two cavities (a) without any coupling, (b) with BS interactions, or (c) with TMS interactions. The distributions are scaled by their maximum value $P_{\max}$. We use $\gamma_1 = \gamma_2$ and $\bar{n}_1 = \bar{n}_2$, and the observation time $t \gg \gamma_1^{-1}$ is fixed so that the mean number of emitted of photons is 25. In panels (a) and (b), we use $\bar{n}_1 = 0.5$, while $\bar{n}_1 = 0.05$ in panel (c). The couplings are $g/\gamma_1 = 1$ in panel (b) and $\lambda/\gamma_1 = 0.2$ in panel (c).

We again recover the expression for two independent cavities by setting $\lambda = 0$. This result is consistent with Ref. [75], which found the cumulant generating function of the total photon current without distinguishing the two cavities (corresponding to $s_1 = s_2$) at zero temperature, $\bar{n}_{1,2} = 0$.

The two cumulant generating functions have a similar analytic structure and both contain a double square root, which was also found in Ref. [76]. Still, they convey different physics. For the TMS interactions, photons are generated by a parametric drive. Hence, $\tilde{K}_{\text{TMS}}$ is non-zero even at zero temperature, whereas the photons in two cavities coupled by BS interactions are created by thermal excitations (or a coherent drive), and hence $\tilde{K}_{\text{BS}} = 0$ at zero temperature. The BS and TMS interactions generate cross-correlations between the emitted photons from the cavities. These correlations can be seen in Fig. 4, where we show the probability distribution $P(n_1, n_2)$ for the number of emitted photons at long times, which we obtain from Eq. (17). For uncoupled cavities, the joint probability in Fig. 4(a) factorizes as $P(n_1, n_2) = P(n_1)P(n_2)$. By constrast, for the BS interactions and TMS interactions in Figs. 4(b,c), we observe clear correlations. Moreover, since the TMS interactions either remove or add a pair of photons — one in each cavity — the correlations are larger than for the BS interactions.

The covariance of emitted photons is found according to Eq. (16). For the sake of simplicity, we assume that the cavities have the same frequency and set $\gamma_1 = \gamma_2$ and $\bar{n}_1 = \bar{n}_2$. We then find

$$\text{Cov}_{\text{BS}}(n_1, n_2) = \gamma_1 t \frac{4g^2}{\gamma_1^2 + 4g^2}(\bar{n}_1 + 1)^2 \bar{n}_1^2, \tag{62}$$

and

$$\text{Cov}_{\text{TMS}}(n_1, n_2) = \gamma_1 t \frac{4\lambda^2}{\gamma_1^2 - 4\lambda^2}(\bar{n}_1 + 1)^2 \left[ \frac{\lambda^2}{\gamma_1^2}(\bar{n}_1 + \bar{n}_A)^2 + \frac{\bar{n}_A}{4}(\bar{n}_A + 2\bar{n}_1) + \frac{\gamma_1^2}{8\lambda^2}\bar{n}_A^2 \right], \tag{63}$$

where the number $\bar{n}_A$ of added photons is defined below Eq. (52). From these expressions, we see that the covariance for BS interactions is smaller than the variance for a single thermal cavity, which reads $\gamma_1 t(\bar{n}_1 + 1)^2 \bar{n}_1^2$ [39]. That is not always the case for TMS interactions.

Figure 5 shows the second cumulants for two identical cavities as a function of the temperature given by the Bose-Einstein factors $\bar{n}_1 = \bar{n}_2$. We show the Fano factors, which are given

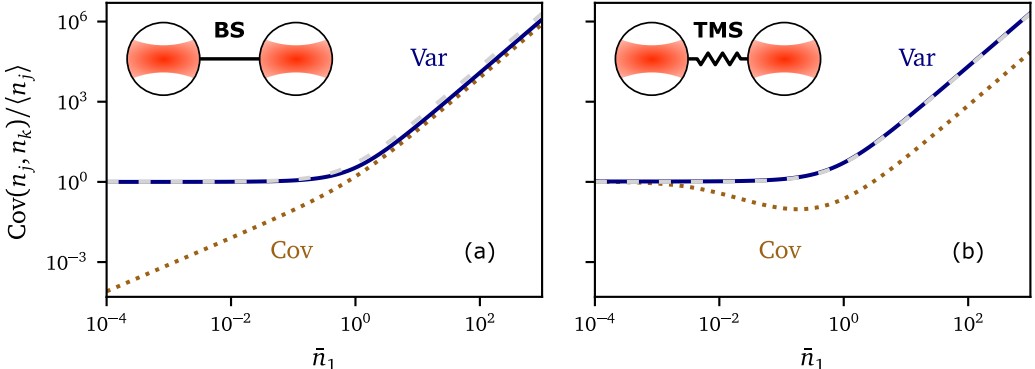

Figure 5: Fano factors. We show the ratio of the (co-)variances over the mean value of emitted photons for (a) BS and (b) TMS interactions. The gray dashed line shows the variance over the mean for a single cavity. Here, the cavities are identical with $\gamma_1 = \gamma_2$, $\bar{n}_1 = \bar{n}_2$, and $g/\gamma_1 = 1$ (BS coupling), or $\lambda/\gamma_1 = 0.05$ (TMS coupling).

by the second cumulants over the average values, and which do not depend on time. At low temperatures, the variance of emitted photons is close to the mean value, indicating that the emission statistics are Poissonian. The Fano factors increase at higher temperatures, but are similar to the variance of a single thermal cavity (gray dashed line), both for BS and TMS interactions. The covariances, by contrast, behave differently for the two types of interactions. For example, at low temperatures, the covariance BS interaction is much smaller than the mean value, whereas for TMS interactions, the covariance approaches the mean value, indicating that emissions from the two cavities are correlated.

Finally, a coherent drive can be included as an additional term in the cumulant generating function following from Eq. (45). Writing the amplitude and the phase of the drive as $f_i = |f_i|e^{i\phi_i}$, we find to lowest order in the coupling, $g, \lambda \ll \sqrt{\gamma_1 \gamma_2}$, the expressions

$$\tilde{K}_{\text{BS}}^{\text{drive}}(\vec{s}) \simeq \sum_{i=1}^{2}\left(\frac{4|f_i|^2\kappa_i}{\gamma_i(\gamma_i - 4\bar{n}_i\kappa_i)}\right) - \frac{16|f_1 f_2|(\kappa_1/\gamma_1 - \kappa_2/\gamma_2)\sin(\phi_1 - \phi_2)}{(\gamma_1 - 4\bar{n}_1\kappa_1)(\gamma_2 - 4\bar{n}_2\kappa_2)}g, \qquad (64)$$

and

$$\tilde{K}_{\text{TMS}}^{\text{drive}}(\vec{s}) \simeq \sum_{i=1}^{2}\left(\frac{4|f_i|^2\kappa_i}{\gamma_i(\gamma_i - 4\bar{n}_i\kappa_i)}\right) - \frac{16|f_1 f_2|(\kappa_1/\gamma_1 + \kappa_2/\gamma_2)\sin(\phi_1 + \phi_2)}{(\gamma_1 - 4\bar{n}_1\kappa_1)(\gamma_2 - 4\bar{n}_2\kappa_2)}\lambda, \qquad (65)$$

where $\kappa_i = \gamma_i(\bar{n}_i + 1)(e^{s_i} - 1)$ now contains the counting fields. In the last terms, we see that the phase difference between the coherent drives is important for the photon emission statistics. Moreover, at low temperatures, the coherent drives dominate the statistics, which are described by two independent Poisson processes that can be controlled by the drives. Because of the interactions between the cavities, we also see that even of only one of the cavities is driven, it will affect the photon emissions from both cavities, see also Eqs. (A.34, A.35) in App. A.4.

## 4.2 Entanglement witness

The photons emitted from the bosonic network carry information about the cavities. In particular, the statistics of emitted photons may be used to characterize the quantum state of the network, at least to some extent. Also, the detection of emitted photons appears to be less disruptive as compared to dispersive measurements, for example, and not as involved as full quantum state tomography. In the context of photon counting statistics, one may use the second-order coherence functions to characterize the entanglement between two photon

sources. However, the coherence functions are of second order in the elements of the covariance matrix. The covariances of the photon emission statistics can also be expressed through time integrals of the second-order coherence functions, and both relate to the zero-frequency noise [65]. Whether the photon emission statistics can be used to detect entanglement for any Gaussian network is an open problem, and here we focus on two cavities with the same frequency that are coupled by TMS and BS interactions, but without a coherent drive.

For Gaussian states, there are several measures of entanglement [58, 77–79]. One of them uses the symplectic eigenvalues, which relate to the negativity [77]. For a two-mode system, the covariance matrix $\Theta$ can be expressed in a block form by using 2-by-2 matrices. The smallest symplectic eigenvalue of the partially transposed covariance matrix can then be written as

$$\nu = \sqrt{\delta - \sqrt{\delta^2 - \det \Theta}}, \tag{66}$$

where

$$\Theta = \begin{pmatrix} \Theta_1 & \Theta_{12} \\ \Theta_{12}^\dagger & \Theta_2 \end{pmatrix}, \tag{67}$$

and

$$\delta = \frac{1}{2}(\det \Theta_1 + \det \Theta_2) - \det \Theta_{12}. \tag{68}$$

Now, the negativity $\mathcal{N} = (\frac{1}{2} - \nu)/2\nu$ is an entanglement monotone, and the bound $\mathcal{N} < 0$ provides a necessary and sufficient condition for separability. In the following, we use the negativity to determine whether or not the two cavities are entangled.

To connect the entanglement to the photon counting statistics, we make use of the Duan criterion, which states that if a quantum state is separable, it holds that [79]

$$D_B = \langle \hat{a}_1^\dagger \hat{a}_1 \rangle + \langle \hat{a}_2^\dagger \hat{a}_2 \rangle - 2|\langle \hat{a}_1 \hat{a}_2 \rangle| \geq 0. \tag{69}$$

This bound can be used as an entanglement witness, since whenever the bound is violated, the state must be entangled. However, there may be entangled states that do not violate the bound. This issue can be mended by using the stronger bound of Ref. [79], but the resulting expression is then nonlinear in the elements of the covariance matrix.

In the language of Gaussian states, the Duan parameter can be expressed as

$$D_B = [\Theta_0]_{11} + [\Theta_0]_{33} - 1 - 2|[\Theta_0]_{14}|, \tag{70}$$

which will be useful in the following, when we relate the Duan parameter to the photon-emission statistics at long times. To this end, we use formal expressions for the mean values and (co-)variances of the photon-emission statistics expressed in terms of the steady-state covariance matrix. For TMS interactions, we can then identify a quantity, which is directly proportional to the Duan parameter, and which reads

$$\begin{aligned}
C_E &= \frac{\mathrm{Var}(n_1)/t - J_1}{(\bar{n}_1 + 1)^2} + \frac{\mathrm{Var}(n_2)/t - J_2}{(\bar{n}_2 + 1)^2} - 2\frac{\mathrm{Cov}(n_1, n_2)/t}{(\bar{n}_1 + 1)(\bar{n}_2 + 1)} \\
&\quad - \frac{\gamma_1 + \gamma_2}{2}\left(\frac{J_1}{\gamma_1(\bar{n}_1 + 1)} - \frac{J_2}{\gamma_2(\bar{n}_2 + 1)}\right)^2 \\
&\quad - (\gamma_1 - \gamma_2)\left(\frac{J_1^2}{\gamma_1^2(\bar{n}_1 + 1)^2} - \frac{J_2^2}{\gamma_2^2(\bar{n}_2 + 1)^2}\right).
\end{aligned} \tag{71}$$

Inserting the mean photon currents $J_{1,2}$ and the (co-)variances at long times, we then find

$$\begin{aligned}
C_E &= \frac{\gamma_1 + \gamma_2}{2}\Big[([\Theta_0]_{11} + [\Theta_0]_{33} - 1)^2 - 4|[\Theta_0]_{14}|^2\Big] \\
&= \frac{\gamma_1 + \gamma_2}{2}([\Theta_0]_{11} + [\Theta_0]_{33} + 2|[\Theta_0]_{14}| - 1)D_B.
\end{aligned} \tag{72}$$

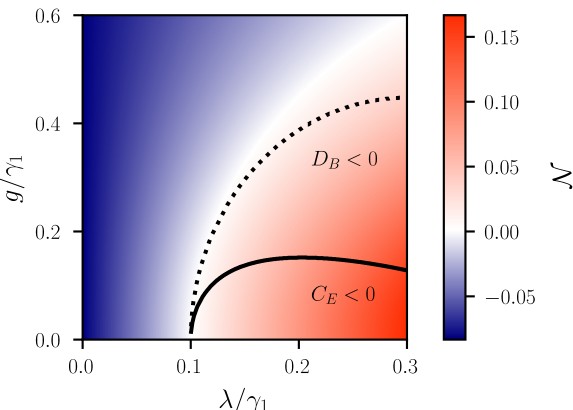

Figure 6: Entanglement between cavities. The negativity as a function of the BS and TMS interaction strengths. Red indicates that the cavities are entangled, while blue corresponds to a separable state. The dotted line shows where the Duan parameter vanishes, while the solid line shows where the parameter derived from the photon emission statistics vanishes. Parameters are $\gamma_1 = \gamma_2$ and $\bar{n}_1 = \bar{n}_2 = 0.1$, and the behavior is similar for other values.

Since the prefactor multiplying $D_B$ is always positive, we can use $C_E$ as a witness of entanglement between two cavities coupled by TMS interactions. Indeed, if the state of the cavities is separable, we have $C_E, D_B \geq 0$. Conversely, if the bound is violated, the cavities must be entangled.

We stress that the considerations above hold for TMS interactions only. Still, it is instructive to investigate if the bound may also provide signatures of entanglement, if BS interactions are included. To this end, we show in Fig. 6 the negativity as a function of the BS and TMS interaction strengths. We also indicate the region where the Duan criterion is violated, $D_B < 0$, and well as the region with $C_E < 0$, which indicates that the two cavities are entangled, if there are no BS interactions. The figures indicate that $C_E < 0$ may also be a signature of entanglement, if the BS interactions are weak, however, we do not have a formal proof of this conjecture. However, from the negativity, we see that increasing BS interactions make the state less entangled, and the correlations produced by the BS interactions appear to be classical in nature. Without BS interactions, $g = 0$, the negativity, the Duan bound, and the condition based on the photon emission statistics give the same separability criterion because of the choice of identical environments for the cavities with $\bar{n}_1 = \bar{n}_2$ and $\gamma_1 = \gamma_2$. If this assumption is lifted, the sufficient and necessary separability conditions differ, and the dashed and full lines in Fig. 6 move to the right. Moreover, if the cavity frequencies differ, the separability condition $C_E \geq 0$ fails for some values of $g$ and $\lambda$. We also note that for two cavities with $\bar{n}_1 = \bar{n}_2$ and $\gamma_1 = \gamma_2$, as in Fig. 6, the entanglement condition $C_E < 0$ can be rewritten in the simple form

$$\mathrm{Var}(n_1 - n_2) < \mathrm{E}(n_1 + n_2), \tag{73}$$

where $\mathrm{E}(\cdot)$ denotes the expectation value. In this form, the entanglement witness seems closely related to a description of nonclassical photon states in quantum optics [50, 80], with the quantum mechanical number operators $\hat{a}_i^\dagger \hat{a}_i$ replaced by the photon emission numbers. Since the photon emission statistics are related to the state of the cavities, this connection might have been expected to a degree. However, the open quantum system description becomes relevant, if the two cavities couple differently to their environments, or if the temperatures are different.

## 4.3 Three-mode circulator

As our last application, we consider a bosonic network consisting of three cavities that are all mutually coupled by BS interactions as shown in Fig. 7(a). This network may function as a bosonic circulator that works without an external magnetic field [16, 81]. In particular, a signal impinging on the first cavity, which is equivalent to a coherent drive, should ideally be transmitted only to the third cavity, while a signal on the second cavity should be transmitted to the first cavity only. This kind of nonreciprocity requires us to break the time-reversal symmetry.

We describe the bosonic circulator by the dynamical matrix

$$\mathcal{A} = \begin{pmatrix} -\frac{\gamma_1}{2}I_2 & \mathcal{C}_{12} & \mathcal{C}_{13} \\ -\mathcal{C}_{12}^\dagger & -\frac{\gamma_2}{2}I_2 & \mathcal{C}_{23} \\ -\mathcal{C}_{13}^\dagger & -\mathcal{C}_{23}^\dagger & -\frac{\gamma_3}{2}I_2 \end{pmatrix}, \qquad \mathcal{C}_{ij} = \begin{pmatrix} ig_{ij} & 0 \\ 0 & -ig_{ij}^* \end{pmatrix}, \tag{74}$$

where $g_{12}, g_{13}$, and $g_{23}$ are the BS interaction constants. In this case, they cannot all be made real by a $U(1)$-transformation, i.e., by shifting the phase of the bosonic operators as $\hat{a}_i \to \hat{a}_i e^{i\phi_i}$. As such, the network breaks the time-reversal symmetry [81]. However, there is a single phase degree of freedom, such that we can write $g_{12} \to g_{12}e^{i\Phi}$ and assume that the couplings, $g_{ij}$, and the phase, $\Phi$, are all real. Due to its role in the breaking of time-reversal symmetry, we refer to $\Phi$ as a synthetic flux. In the simplest case, where $\gamma_i = \gamma$ and $g_{ij} = g$, the condition for ideal circulation reads $g = \gamma/2$ and $\Phi = \pm\frac{\pi}{2}$, and we note that more complex conditions can be formulated, if the couplings or the dissipation rates are not all the same.

In practice, the accurate control of the interaction strengths and the dissipation rates is difficult, although some progress has recently been made [82]. Also, in Ref. [17], the vibrational modes of a mechanical resonator were coupled, and the interactions were controlled using an optomechanical coupling combined with an external drive. It should also be noted that even a non-ideal circulator displays interesting physics, which affect the photon counting statistics.

In the following, we evaluate the photon emission statistics using the methods of Sec. 3.4. We first take the temperature to be zero and assume that a coherent drive is applied only to the first cavity. In that case, the scaled cumulant generating function becomes

$$\tilde{K}(\vec{s}) = p\Big[(4g^2 + \gamma^2)^2(e^{s_1} - 1) + 4g^2(4g^2 + \gamma^2 - 4g\gamma\sin\Phi)(e^{s_2} - 1) \\ + 4g^2(4g^2 + \gamma^2 + 4g\gamma\sin\Phi)(e^{s_3} - 1)\Big], \tag{75}$$

where the prefactor reads

$$p = 4\gamma|f_1|^2/[(8g^2 + \gamma^2)(16g^4 + 16g^2\gamma^2 + \gamma^4) + 128g^6\cos 2\Phi]. \tag{76}$$

We then see that the photon emission statistics is a combination of three independent Poisson processes. Moreover, the coherent drive on the first cavity leads to photon emissions from all cavities. However, for the special case of $\Phi = \frac{\pi}{2}$ and $g = \gamma/2$, we find the simple expression

$$\tilde{K}(\vec{s}) = |f_1|^2(e^{s_1} - 1 + e^{s_3} - 1)/\gamma, \tag{77}$$

showing that the photon emissions become evenly distribution between the first and the third cavity only, while no photons are emitted from the second cavity.

Next, we investigate how a finite temperature affects the emission of photons that are created by the coherent drive. Focusing on the second cavity, we find for three identical cavities

$$\tilde{K}_{\text{drive}}(s_2, s_{1,3} = 0) = \frac{4|f_1|^2}{\gamma}\frac{(1 - \sin\Phi)(\bar{n} + 1)(e^{s_2} - 1)}{9 + \cos 2\Phi - 16\bar{n}(\bar{n} + 1)(e^{s_2} - 1)}, \tag{78}$$

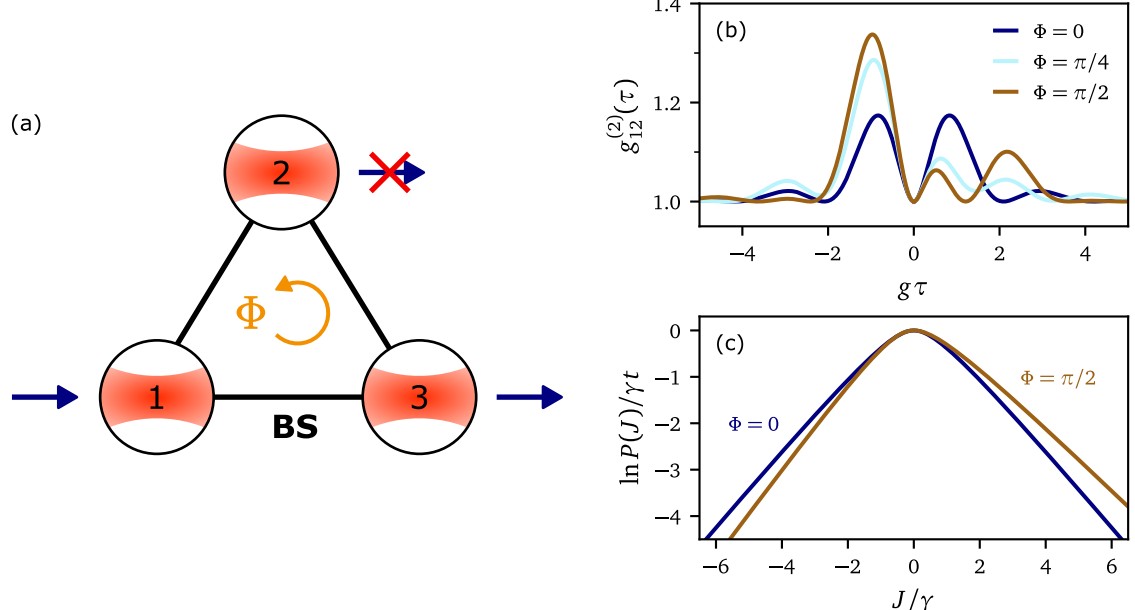

Figure 7: Bosonic circulator. (a) The circulator consists of three coupled cavities. An input signal on the first cavity is ideally transmitted to the third cavity only. The direction of circulation depends on the synthetic flux $\Phi$. (b) Second-order cross-coherence function for different values of $\Phi$ with $g = \gamma$. (c) Large-deviation statistics of the net currents for a circulator with $g = \gamma/2$. The current in the first and second cavity are fixed to be the opposite, $J = J_1^{\text{net}} = -J_2^{\text{net}}$. The three cavities are non-driven and have the same temperature, $\bar{n}_{1,2,3} = 1$, so that the average currents vanish.

where we again take $g = \gamma/2$, and we have left out a term that describes the emission of thermal photons in the absence of the drive. From this expression, we see that a finite temperature leads to photon emissions that are not Poissonian. However, ideal circulation can still be achieved for $\Phi = \frac{\pi}{2}$, where no photons due to the drive are emitted from the second cavity.

Interestingly, the physics of photon circulation also appears in the cross-correlations of the thermal photons without the coherent drive. In Fig. 7(b), we show the coherence function of emitting a photon from the first cavity followed by an emission from the second cavity. With a finite synthetic flux, the coherence function is not symmetric in time, and it is more likely that the second cavity emits a photon before the first cavity and not the other way around.

Finally, we consider the distribution of the net photon currents in the circulator. To this end, we set $\vec{u} = -\vec{s}$, so that we count the net flow of photons out of the cavities as $\vec{n} - \vec{m}$, where $\vec{n}$ and $\vec{m}$ are the number of emitted and absorbed photons, respectively. We then solve the photon counting equations for the cumulant generating function. At long times, the evaluation of the probability distribution is simplified by a saddle-point approximation of the integral in Eq. (17). The large-deviation statistics of the net photon currents then becomes [44]

$$\ln(P)/t \simeq \tilde{K}(\vec{s}_{\text{sp}}) - \vec{s}_{\text{sp}} \cdot \vec{J}_{\text{net}}, \tag{79}$$

where $\vec{J}_{\text{net}} = (\vec{n} - \vec{m})/t$ are the net currents, and $\vec{s}_{\text{sp}}$ solves the saddle-point equation

$$\nabla_{\vec{s}} \tilde{K}(\vec{s}_{\text{sp}}) = \vec{J}_{\text{net}}. \tag{80}$$

We solve the saddle-point equation numerically by minimizing the function

$$f(\vec{s}) = \left| \nabla_{\vec{s}} \tilde{K}(\vec{s}) - \vec{J}_{\text{net}} \right|.$$

To this end, we find the cumulant generating function at long times by solving the Riccati equation (25a), and we evaluate the partial derivatives using a finite difference method.[1] In Fig. 7(c), we show the statistics of the net photon currents for the first two cavities with thermal environments, having set $J_1^{\text{net}} = -J_2^{\text{net}} = J$. For an ideal circulator with $g = \gamma/2$ and $\Phi = \pi/2$, there is an asymmetry in the photon currents. It is more likely that photons flow out of the first cavity ($J_1^{\text{net}} > 0$), and photons flow into the second cavity ($J_2^{\text{net}} < 0$), than without the symmetry-breaking synthetic flux. Also, if the photons circulate as in Fig. 7(a) and $g < \gamma$, thermal photons that are absorbed by the second cavity may be transferred to the first cavity and emitted there. By contrast, the flow in the other direction is blocked, and photons only flow from the first to the third cavity.

## 5 Conclusions

We have developed a comprehensive theoretical framework to describe the photon counting statistics of Gaussian bosonic networks. Such networks may consist of microwave cavities that are coherently driven by external light fields and coupled by beam splitter interactions and two-mode squeezing. However, our theoretical framework can equally well be applied to other types of coupled modes such as nanomechanical resonators in the quantum regime. Our approach is based on quantum mechanical phase-space methods and the use of Gaussian states, which allow us to describe the state of the bosonic network by a displacement vector and a covariance matrix only. The dynamics of the covariance matrix is governed by a Lyapunov equation, which generalizes to a Riccati equation, when counting fields are included. By solving the Riccati equation, we can evaluate the photon counting statistics both at finite times and at long times. At finite times, the distribution of waiting times and the second-order coherence functions are of particular interest, since they contain information about the temporal correlations between photon emissions. At long times, we can evaluate the average photon currents and their fluctuations, which are encoded in the cumulants of the photon counting statistics. It is also possible to consider the large-deviation statistics of the photon currents.

To illustrate our theoretical framework, we have considered three specific applications, focusing on small bosonic networks, which are of relevance to recent experiments. However, our formalism also applies to larger networks, and the dimension of the involved matrix equations grows only linearly with the number of cavities, which for practical purposes makes them solvable. For two coupled cavities, we have presented a detailed investigation of the photon counting statistics, including the distribution of waiting times, the second-order coherence functions, and the photon emission statistics at long times, with an emphasis on the cross-correlations between the emitted photons. Building on this setup, we have discussed the entanglement between the cavities and how it may be detected from measurements of the photon counting statistics. In particular, for two cavities coupled by two-mode squeezing, the first and second cumulants of the photon counting statistics can be combined to form a quantity, which is directly proportional to the Duan parameter for continuous variable systems and therefore may function as an entanglement witness. We have also found indications that it can detect entanglement, if beamsplitter interactions are included. Finally, we have considered a bosonic circulator consisting of three mutually coupled microwave cavities with a synthetic flux that breaks the time-reversal symmetry. For this system, the parameters can be chosen so that an incoming signal on the first cavity is transmitted to the third cavity but not the second. We have evaluated the photon counting statistics of the circulator at long times and analyzed how the non-reciprocal properties affect the cross-correlations and the large-deviation statistics.

---

[1]The Python-based scripts to generate our figures are available at https://doi.org/10.5281/zenodo.12773744.

Our theoretical framework can be applied to a variety of systems, and our work can be extended in many different directions. For example, it would be interesting to explore the heat fluctuations in chains of quantum harmonic oscillators [83]. Our approach may also be used to investigate the transport statistics in other bosonic systems, such as heat diodes and heat engines, in the context of quantum thermodynamics [35]. Moreover, with the development of efficient detectors of single microwave photons, our predictions may be tested in future experiments [22,23,27]. On top of these applications, there are several possible avenues for further developments. While we have discussed the entanglement between two coupled cavities, it would be interesting to understand, if genuine multipartite entanglement in a large bosonic network can be detected from measurements of the photon counting statistics [84]. Also, while we have focused on the statistics of photon counts in a fixed time interval, one could instead consider the first-passage time distributions and investigate how long time it takes until a certain number of photons have been emitted or absorbed [85–87]. Finally, it may be possible to extend our framework to non-Gaussian systems with Kerr or other non-linearities [4]. Such situations would include hybrid networks that couple fermionic and bosonic degrees of freedom. One can imagine hybrid networks consisting of microwave cavities, functioning as connectors, which are coupled to the electron transport in quantum dots [29, 88, 89] or Josephson junctions [90,91], functioning as nodes. On top of this, one may consider explicitly time-dependent situations, where the frequencies of the modes are modulated in time [92].

# Acknowledgments

**Funding information**   We acknowledge the support from the Research Council of Finland through the Finnish Centre of Excellence in Quantum Technology (Grant No. 352925) and the Japan Society for the Promotion of Science through an Invitational Fellowship for Research in Japan. This work was partially supported by the Wallenberg Centre for Quantum Technology (WACQT) funded by Knut and Alice Wallenberg Foundation.

# A   Appendices

## A.1   Photon counting equations

Here, we transform the generalized master equation from the density matrix $\hat{\rho}$ to the characteristic function $\chi$. To this end, we map the bosonic operators to differential operators in phase space. These mappings are straightforward to derive by differentiating the displacement operator $D(\boldsymbol{\alpha}) = \exp(\hat{\boldsymbol{a}}^{\dagger}\mathcal{K}\boldsymbol{\alpha})$, when ordered either normally or antinormally, using the Baker–Hausdorff–Campbell lemma [59–61]. We then arrive at the four relations

$$\hat{a}_j D(\boldsymbol{\alpha}) = \left( \frac{\alpha_j}{2} - \frac{\partial}{\partial \alpha_j^*} \right) D(\boldsymbol{\alpha}),$$

$$\hat{a}_j^{\dagger} D(\boldsymbol{\alpha}) = \left( \frac{\alpha_j^*}{2} + \frac{\partial}{\partial \alpha_j} \right) D(\boldsymbol{\alpha}),$$

$$D(\boldsymbol{\alpha})\hat{a}_j = -\left( \frac{\alpha_j}{2} + \frac{\partial}{\partial \alpha_j^*} \right) D(\boldsymbol{\alpha}),$$

$$D(\boldsymbol{\alpha})\hat{a}_j^{\dagger} = -\left( \frac{\alpha_j^*}{2} - \frac{\partial}{\partial \alpha_j} \right) D(\boldsymbol{\alpha}).$$

(A.1)

Equation (21) for the characteristic function $\chi$ resembles a Fokker–Planck equation, and it can be derived by multiplying the generalized master equation (18) with the displacement operator, taking the trace, and using the cyclical trace properties together with the relations above.

Next, we use the Gaussian Ansatz of the main text to derive the photon counting statistics. First, we set $\chi = \exp(K - Q)$, where $K$ is the cumulant generating function, and we define

$$Q = \frac{1}{2}\boldsymbol{\alpha}^\dagger \mathcal{K}\Theta\mathcal{K}\boldsymbol{\alpha} - \boldsymbol{d}^\dagger \mathcal{K}\boldsymbol{\alpha}, \tag{A.2}$$

where the matrix $\mathcal{K} = I_N \otimes \sigma_z$ is given by a Kronecker product, and $\sigma_z$ is the third Pauli matrix. Due to the exponential structure of the Ansatz, Eq. (21) leads to the expression

$$\frac{\mathrm{d}K}{\mathrm{d}t} - \frac{\mathrm{d}Q}{\mathrm{d}t} = -\left\{ i\boldsymbol{\alpha}^T \mathcal{H}^T \mathcal{K} \frac{\partial}{\partial\boldsymbol{\alpha}} + \sum_{j=1}^N \gamma_j \left[ B_j\left(\alpha_j \frac{\partial}{\partial\alpha_j} + \alpha_j^* \frac{\partial}{\partial\alpha_j^*}\right) + C_j \frac{\partial^2}{\partial\alpha_j\partial\alpha_j^*} \right] \right\} Q$$
$$- \boldsymbol{f}^\dagger \boldsymbol{\alpha} + \sum_{j=1}^N \gamma_j \left( A_j + C_j \frac{\partial Q}{\partial\alpha_j} \frac{\partial Q}{\partial\alpha_j^*}\right), \tag{A.3}$$

since $K$ does not depend on $\boldsymbol{\alpha}$. There are terms of different orders in $\boldsymbol{\alpha}$, which must be matched on the left- and right-hand sides. First, we note that the variables $B_j$ and $C_j$ do not depend on $\boldsymbol{\alpha}$. On the other hand, the derivatives of $Q$ with respect to $\boldsymbol{\alpha}$ can be written as

$$\frac{\partial Q}{\partial\alpha_j} = \mathbf{e}_{2j}^T \mathcal{K}\Theta\mathcal{K}\boldsymbol{\alpha} + \boldsymbol{d}^\dagger \mathcal{K}\mathbf{e}_{2j-1}, \tag{A.4}$$

$$\frac{\partial Q}{\partial\alpha_j^*} = \mathbf{e}_{2j-1}^T \mathcal{K}[\Theta\mathcal{K}\boldsymbol{\alpha} - \boldsymbol{d}] = \boldsymbol{\alpha}^\dagger \mathcal{K}\Theta\mathcal{K}\mathbf{e}_{2j} + \boldsymbol{d}^\dagger \mathcal{K}\mathbf{e}_{2j}, \tag{A.5}$$

where $\mathbf{e}_k$ is a unit vector with only the $k$'th element being one, all others are zero. These relations rely on the block structure of $\Theta$ with $[\Theta]_{2j-1,2j-1} = [\Theta]_{2j,2j}$ and $[\boldsymbol{d}]_{2j} = [\boldsymbol{d}]_{2j-1}^*$.

Next, we define the matrix $\mathcal{S} = I_N \otimes \sigma_x$, and we then have $\boldsymbol{\alpha}^* = \mathcal{S}\boldsymbol{\alpha}$ and $\mathcal{S}\Theta\mathcal{S} = \Theta^* = \Theta^T$. The matrices $\mathcal{S}$ and $\mathcal{K}$ inherit the anticommutation relations and involutory property of Pauli matrices, since $\mathcal{S}\mathcal{K} = -\mathcal{K}\mathcal{S}$ and $\mathcal{S}^2 = \mathcal{K}^2 = I_{2N}$. In addition, it is useful that one can write $\boldsymbol{x}_1^T Y \boldsymbol{x}_2 = \boldsymbol{x}_2^T Y^T \boldsymbol{x}_1$ for any two vectors $\boldsymbol{x}_{1,2}$ and any matrix $Y$, and we can also write

$$\sum_j \gamma_j A_j = -\frac{1}{2}\operatorname{tr}(\Gamma_s - \Gamma_u) - \frac{1}{2}\boldsymbol{\alpha}^\dagger \mathcal{Q}\boldsymbol{\alpha}. \tag{A.6}$$

After evaluating the derivatives and collecting terms to the same power in $\boldsymbol{\alpha}$, we arrive at

$$\frac{\mathrm{d}K}{\mathrm{d}t} = -\sum_j \gamma_j C_j\left(\frac{1}{2}\mathbf{e}_{2j}^T \mathcal{K}\Theta\mathcal{K}\mathbf{e}_{2j} + \frac{1}{2}\mathbf{e}_{2j-1}^T \mathcal{K}\Theta\mathcal{K}\mathbf{e}_{2j-1} + \boldsymbol{d}^\dagger \mathcal{K}\mathcal{E}_j\mathcal{K}\boldsymbol{d}\right) - \frac{1}{2}\operatorname{tr}(\Gamma_s - \Gamma_u), \tag{A.7}$$

$$\frac{\mathrm{d}\boldsymbol{d}^\dagger}{\mathrm{d}t}\mathcal{K}\boldsymbol{\alpha} = -i\boldsymbol{d}^\dagger \mathcal{H}\boldsymbol{\alpha} + \sum_j \gamma_j B_j \boldsymbol{d}^\dagger \mathcal{K}\boldsymbol{\alpha} - \sum_j \gamma_j C_j \boldsymbol{d}^\dagger \mathcal{K}\mathcal{E}_j\mathcal{K}\Theta\mathcal{K}\boldsymbol{\alpha} - \boldsymbol{f}^\dagger \boldsymbol{\alpha}, \tag{A.8}$$

and

$$\frac{1}{2}\boldsymbol{\alpha}^\dagger \mathcal{K}\frac{\mathrm{d}\Theta}{\mathrm{d}t}\mathcal{K}\boldsymbol{\alpha} = -\sum_j \gamma_j C_j \boldsymbol{\alpha}^\dagger \mathcal{K}\Theta\mathcal{K}\mathcal{E}_j\mathcal{K}\Theta\mathcal{K}\boldsymbol{\alpha} - i\boldsymbol{\alpha}^\dagger \mathcal{H}\Theta\mathcal{K}\boldsymbol{\alpha} + \frac{1}{2}\boldsymbol{\alpha}^\dagger \mathcal{Q}\boldsymbol{\alpha}, \tag{A.9}$$

where we have defined $\mathcal{E}_j = \frac{1}{2}(\mathbf{e}_{2j}\mathbf{e}_{2j}^T + \mathbf{e}_{2j-1}\mathbf{e}_{2j-1}^T)$. We could also have simplified the expressions so that $\mathcal{E}_j$ would become any linear combination of $\mathbf{e}_{2j}\mathbf{e}_{2j}^T$ and $\mathbf{e}_{2j-1}\mathbf{e}_{2j-1}^T$ and similarly

for the first term in the equation for $dK/dt$. That would be possible because of the block structure of the covariance matrix $\Theta$. Here, we chose the symmetric version so that this block structure is conserved in the time evolution of the generalized covariance matrix $\Theta(t;\vec{s};\vec{u})$.

Finally, we remove the inner product structure with $\boldsymbol{\alpha}$ and simplify the resulting equations. As a detail, one should note that $\Theta$ should be a Hermitian matrix at all times, and hence the Hamiltonian term $-i\boldsymbol{\alpha}^\dagger \mathcal{H}\Theta\mathcal{K}\boldsymbol{\alpha}$ must be equal to its Hermitian conjugate. Noting that $\sum_j \gamma_j C_j \mathcal{E}_j = -\Gamma_s - \Gamma_u$ and $\mathcal{K}\Gamma_{s,u}\mathcal{K} = \Gamma_{s,u}$, we then find the photon counting equations (25).

## A.2 Other phase-space representations

Here, we briefly discuss the Wigner function and other phase-space representations, which can be used to put Eq. (21) into different forms which may simplify the photon counting problem. The Wigner function can be expressed using a Fourier transformation as

$$W(\boldsymbol{\alpha}) = \frac{1}{\pi^{2N}} \int_{-\infty}^{\infty} d\boldsymbol{\xi}\, \chi(\boldsymbol{\xi}) \exp(\boldsymbol{\xi}^\dagger \mathcal{K}\boldsymbol{\alpha}), \tag{A.10}$$

where the integration measure is $d\boldsymbol{\xi} = \prod_{j=1}^{N} d\mathrm{Re}(\xi_j)\, d\mathrm{Im}(\xi_j)$ and $\boldsymbol{\xi} = (\xi_1, \xi_1^*, \dots)^T$ [59–61]. The factor $\pi^{2N}$ ensures that an integral of $W(\boldsymbol{\alpha})$ with the same measure gives unity. Using the connection between $W$ and $\chi$, we may derive the corresponding equation for the Wigner function from Eq. (21). To this end, we first express the function $A_j$ as

$$A_j = -\frac{1}{2}\big[(\bar{n}_j + 1)(e^{s_j} - 1) - \bar{n}_j(e^{u_j} - 1)\big] - \left[\left(\bar{n}_j + \frac{1}{2}\right) + \frac{\bar{n}_j + 1}{4}(e^{s_j} - 1) + \frac{\bar{n}_j}{4}(e^{u_j} - 1)\right]|\alpha_j|^2, \tag{A.11}$$

and then split it into two parts as $A_j \equiv A_j^{(0)} + A_j^{(1)}|\alpha_j|^2$. Using the definition (A.10) for the generalized Wigner function $W = W(t;\vec{s},\vec{u};\boldsymbol{\alpha})$ combined with partial integration, we then find

$$\frac{\partial W}{\partial t} = \left\{ i\left(\frac{\partial}{\partial \boldsymbol{\alpha}}\right)^T \mathcal{K}\mathcal{H}\boldsymbol{\alpha} + \boldsymbol{f}^T \mathcal{K}\frac{\partial}{\partial \boldsymbol{\alpha}} - \sum_{j=1}^{N} \gamma_j \left[ A_j^{(1)} \frac{\partial^2}{\partial \alpha_j \partial \alpha_j^*} + B_j\left(\frac{\partial}{\partial \alpha_j}\alpha_j + \frac{\partial}{\partial \alpha_j^*}\alpha_j^*\right)\right]\right\} W$$

$$- \sum_{j=1}^{N} \gamma_j \left(A_j^{(0)} + C_j |\alpha_j|^2\right) W, \tag{A.12}$$

where we have introduced the row vector $(\partial/\partial \boldsymbol{\alpha})^T = (\partial/\partial \alpha_1 \; \partial/\partial \alpha_1^* \dots)$. The terms on the second line vanish without the counting fields. The equation for the Wigner function is equivalent to the result in Ref. [76], although expressed in a different basis. This approach could prove useful to investigate the photon counting statistics of bosonic systems, where Kerr or other non-linearities are involved, which are beyond the Gaussian state description here.

Next, if one only considers photon emissions or absorptions, other phase-space representations may simplify Eq. (21). The $P$ (or Glauber–Sudarshan) distribution and the $Q$ (or Husimi) distribution are related to the normal- and antinormal ordered characteristic functions $\chi_N$ and $\chi_A$, respectively [59–61]. These are related to the characteristic function as $\chi(\boldsymbol{\alpha})$ by $\chi_N(\boldsymbol{\alpha}) = \exp(\boldsymbol{\alpha}^\dagger \boldsymbol{\alpha}/2)\chi(\boldsymbol{\alpha})$ and $\chi_A(\boldsymbol{\alpha}) = \exp(-\boldsymbol{\alpha}^\dagger \boldsymbol{\alpha}/2)\chi(\boldsymbol{\alpha})$. Working in a rotating frame of the Hamiltonian $\hat{H}$, we obtain for $\chi_{N,A} = \chi_{N,A}(t;\vec{s};\boldsymbol{\alpha})$ the equations

$$\frac{d}{dt}\chi_N = -\sum_{j=1}^{N} \gamma_j \left[ \bar{n}_j |\alpha_j|^2 + \frac{1}{2}\left(\alpha_j \frac{\partial}{\partial \alpha_j} + \alpha_j^* \frac{\partial}{\partial \alpha_j^*}\right) + (\bar{n}_j + 1)(e^{s_j} - 1)\frac{\partial^2}{\partial \alpha_j \partial \alpha_j^*}\right]\chi_N, \tag{A.13}$$

and

$$\frac{d}{dt}\chi_A = -\sum_{j=1}^{N} \gamma_j \left[ (\bar{n}_j + 1)|\alpha_j|^2 + \frac{1}{2}\left(\alpha_j \frac{\partial}{\partial \alpha_j} + \alpha_j^* \frac{\partial}{\partial \alpha_j^*}\right) + \bar{n}_j(e^{u_j} - 1)\frac{\partial^2}{\partial \alpha_j \partial \alpha_j^*}\right]\chi_A. \tag{A.14}$$

The property that the counting fields only appear in the last terms is helpful for many analytical calculations. For instance, it simplifies the expansions in the counting fields in App. A.5.

In principle, we can derive the photon counting equations from the equations for $\chi_{N,A}$. However, for Gaussian states, we can find the covariance matrices from the definitions of the normal- and antinormal-ordered characteristic functions, $\Theta_N = \Theta - I_{2N}/2$ and $\Theta_A = \Theta + I_{2N}/2$. We can then insert these covariance matrices into equations (25) to find their time evolution.

Finally, we briefly discuss the equivalence between the phase-space formalism and third quantization [46]. In the former, both density matrices and operators are transformed into complex functions. By contrast, in the latter, both are treated as elements of a vector space. The photon counting equations in phase space may also be formulated in the third quantization formalism [47], allowing for further connections to Keldysh path integral approaches [46]. That may provide another method to evaluate the photon counting statistics of cavity networks.

## A.3 Two-time correlators

Here, we discuss how the correlators of two photon-emissions can be evaluated. First, we write

$$c_{jk}(\tau;\vec{s}) = \mathrm{tr}\big(\mathcal{J}_k e^{\mathcal{L}(\vec{s})\tau} \mathcal{J}_j \hat{\rho}_0\big) = D_k \mathcal{T}(\tau;\vec{s}) D_j \chi_0(\boldsymbol{\alpha})\Big|_{\boldsymbol{\alpha}=0}, \tag{A.15}$$

where the differential operators $D_{j,k}$ directly follow from Eqs. (A.1), and $\mathcal{T}$ is the time evolution operator in the phase-space representation. Since we here consider emission processes only, we work with the normal-ordered covariance matrices $\Theta_N = \Theta - I_{2N}/2$. The main challenge is that the state of the system, given by $\chi$, is not Gaussian right after a photon emission.

Assuming a Gaussian steady state $\chi_0 = \chi_0(\boldsymbol{\alpha})$, we need to time evolve the quantity

$$D_j \chi_0 = \left[\frac{1}{2}\mathrm{tr}\big(\Gamma_j \Theta_{N,0}\big) + \frac{1}{2}\boldsymbol{d}_0^\dagger \Gamma_j \boldsymbol{d}_0 + \boldsymbol{d}_0^\dagger \Gamma_j \Theta_{N,0}\boldsymbol{\alpha} - \frac{1}{2}\boldsymbol{\alpha}^\dagger \Theta_{N,0}\Gamma_j \Theta_{N,0}\boldsymbol{\alpha}\right]\chi_0 \equiv F_j(\boldsymbol{\alpha})\chi_0. \tag{A.16}$$

Now, this unnormalized characteristic function should be inserted into Eq. (21). To this end, we use an Ansatz, where we fix the functional as time evolves. Specifically, we write

$$\mathcal{T}(\tau;\vec{s})D_j \chi_0 = f(t;\boldsymbol{\alpha})\chi_{\mathrm{ans}} \equiv \left[\frac{1}{2}\mathrm{tr}\big(\Gamma_j \Theta_{N,0}\big) + \frac{1}{2}\boldsymbol{d}_0^\dagger \Gamma_j \boldsymbol{d}_0 + z(t) + \boldsymbol{y}^\dagger(t)\boldsymbol{\alpha} - \frac{1}{2}\boldsymbol{\alpha}^\dagger X(t)\boldsymbol{\alpha}\right]\chi_{\mathrm{ans}}, \tag{A.17}$$

where $\chi_{\mathrm{ans}} = \chi_{\mathrm{ans}}(t;\vec{s};\boldsymbol{\alpha})$ is the Ansatz of Eq. (24) that solves Eq. (21). The initial conditions for $z$, $\boldsymbol{y}$, and $X$ are found from $f(0;\boldsymbol{\alpha}) = F_j(\boldsymbol{\alpha})$. Moreover, the polynomial $f$ evolves as

$$\begin{aligned}
\frac{\mathrm{d}f(t;\boldsymbol{\alpha})}{\mathrm{d}t} = &\left[i\boldsymbol{\alpha}^T \mathcal{H}^T \mathcal{K}\frac{\partial}{\partial\boldsymbol{\alpha}} + \sum_{j=1}^N \gamma_j B_j\left(\alpha_j \frac{\partial}{\partial\alpha_j} + \alpha_j^*\frac{\partial}{\partial\alpha_j^*}\right)\right]f(t;\boldsymbol{\alpha}) \\
&+ \sum_{j=1}^N \gamma_j C_j\left(\frac{\partial^2}{\partial\alpha_j\partial\alpha_j^*} + \frac{\partial\ln\chi_{\mathrm{ans}}}{\partial\alpha_j^*}\frac{\partial}{\partial\alpha_j} + \frac{\partial\ln\chi_{\mathrm{ans}}}{\partial\alpha_j}\frac{\partial}{\partial\alpha_j^*}\right)f(t;\boldsymbol{\alpha}),
\end{aligned} \tag{A.18}$$

which we obtain by using the chain rule. After some algebra, as in App. A.1, we then obtain differential equations for $X$, $\boldsymbol{y}$, and $z$ as in Eqs. (41a)–(41c). Finally, by operating on $f(t;\boldsymbol{\alpha})\chi_{\mathrm{ans}}$ with $D_k$ and setting $\boldsymbol{\alpha} = 0$, we arrive at Eq. (39) of the main text.

In many cases, the waiting time distributions are biexponential, and the decay at long times is characterized by two time constants. At long times, we infer from Eqs. (39) and (41) that

$$\ln W_{jk}(\tau) \simeq K(\tau, s_k \to -\infty) \simeq \tau\tilde{K}(s_k \to -\infty), \tag{A.19}$$

such that the decay rate is given by $\gamma_\infty \simeq -\tilde{K}(s_k \to -\infty)$. We can then evaluate the decay rate using the methods in Sec. 3.4. The short-time behavior of the waiting time distribution can be found by solving the differential equations (25) and (41) for a short time-step. Using the initial conditions arising from the thermal state together with the normal-ordered covariance matrix $\Theta_{N,0}$ and the first moments $\boldsymbol{d}_0$, we find for a small time-step $\delta t$ that

$$W_{jk}(\delta t) \simeq W_{jk}(0)e^{-\gamma_0 \delta t}, \tag{A.20}$$

with the decay rate $\gamma_0$ defined by

$$\gamma_0 J_j W_{jk}(0) = J_k J_j W_{jk}(0) + J_j G_{kk} + J_k G_{jk} + \text{tr}\big(\Gamma_k \Theta_{N,0} \Gamma_k \Theta_{N,0} \Gamma_j \Theta_{N,0}\big) \tag{A.21}$$
$$- \text{tr}\left(\frac{\Gamma_k \mathcal{A} + \mathcal{A}^\dagger \Gamma_k}{2} \Theta_{N,0} \Gamma_j \Theta_{N,0}\right) + \boldsymbol{d}_0^\dagger \Gamma_k (2\Theta_{N,0}\Gamma_k - \mathcal{A})\Theta_{N,0}\Gamma_j \boldsymbol{d}_0,$$

where we have defined $W_{jk}(0) = J_k + G_{jk}/J_j$ and $G_{jk} = \frac{1}{2}\text{tr}\big(\Gamma_j \Theta_{N,0}\Gamma_k \Theta_{N,0}\big) + \boldsymbol{d}_0^\dagger \Gamma_j \Theta_{N,0}\Gamma_k \boldsymbol{d}_0$, and $J_k$ is the mean photon emission current from cavity $k$.

## A.4 Long-time statistics of two cavities

Here, we derive the cumulant generating function of photon emissions for the beam-splitter (BS) and the two-mode-squeezing (TMS) interactions at long times using the methods of Sec. 3.4. The calculations are simplified by the use of the normal-ordered covariance matrix whose Riccati equation is given in Eq. (55).

To begin with, we find the matrix $\mathcal{F}$ from the relation $\mathcal{F}\mathcal{F}^\dagger = \mathcal{A}\Gamma_s^{-1}\mathcal{A}^\dagger - \mathcal{B}$, where the $\mathcal{A}$'s for BS and TMS systems are given in Eq. (49). For TMS interactions, we have

$$\mathcal{B}_{\text{TMS}} = \begin{pmatrix} \gamma_1 \bar{n}_1 & 0 & 0 & -i\lambda \\ 0 & \gamma_1 \bar{n}_1 & i\lambda & 0 \\ 0 & -i\lambda & \gamma_2 \bar{n}_2 & 0 \\ i\lambda & 0 & 0 & \gamma_2 \bar{n}_2 \end{pmatrix}, \tag{A.22}$$

while for the BS interactions, we have $\mathcal{B}_{\text{BS}} = \bigoplus_{j=1}^2 \gamma_j \bar{n}_j I_2$. Having found $\mathcal{F}$, we insert it into the expression $\Theta_N(\vec{s}) = \mathcal{F}\Gamma_s^{-1/2} - \mathcal{A}\Gamma_s^{-1}$ and find the scaled CGF at long times, $\tilde{K}(\vec{s}) = \frac{1}{2}\text{tr}[\Gamma_s \Theta_N(\vec{s})]$, without the coherent drive. Finally, we use Eq. (45) (with the replacements $\mathcal{W} \to \mathcal{A}$ and $\Gamma_u = 0$) to find the contribution from the drive. In practice, we only have to solve for the diagonal elements of $\mathcal{F}$, since $\Gamma_s$ is diagonal. Moreover, we denote $\gamma_i(\bar{n}_i + 1)(e^{s_i} - 1) \equiv \kappa_i$ in $\Gamma_s$. The calculation for the BS and TMS interactions are nearly identical, and we explain the process for the former in detail, while we only point out the main differences for the latter.

Based on the matrix structure of $\mathcal{F}\mathcal{F}^\dagger = \mathcal{A}_{\text{BS}}\Gamma_s^{-1}\mathcal{A}_{\text{BS}}^\dagger - \mathcal{B}_{\text{BS}}$, we use the Ansatz

$$\mathcal{F} = \begin{pmatrix} F_1 & 0 & F_3 & 0 \\ 0 & F_1^* & 0 & F_3^* \\ F_4 & 0 & F_2 & 0 \\ 0 & F_4^* & 0 & F_2^* \end{pmatrix}. \tag{A.23}$$

With this Ansatz, we only have to determined four complex numbers, rather than sixteen.

Now, noting that $\Theta_N$ should be Hermitian for real-valued counting fields and recalling that $\Theta_N(\vec{s}) = \mathcal{F}\Gamma_s^{-1/2} - \mathcal{A}\Gamma_s^{-1}$, we find $\text{Im} F_{1,2} = 0$ together with the two linear relations

$$\frac{\text{Re} F_3}{\sqrt{\kappa_2}} = \frac{\text{Re} F_4}{\sqrt{\kappa_1}}, \quad \frac{\text{Im} F_3}{\sqrt{\kappa_2}} + \frac{\text{Im} F_4}{\sqrt{\kappa_1}} = -g\left(\frac{1}{\kappa_1} + \frac{1}{\kappa_2}\right). \tag{A.24}$$

In addition, the expression $\mathcal{F}\mathcal{F}^\dagger = \mathcal{A}_{\text{BS}}\Gamma_s^{-1}\mathcal{A}_{\text{BS}}^\dagger - \mathcal{B}_{\text{BS}}$ yields four quadratic equations

$$|F_1|^2 + |F_3|^2 = \frac{g^2}{\kappa_2} + \frac{\gamma_1^2/4}{\kappa_1} - \gamma_1\bar{n}_1\,,$$

$$|F_2|^2 + |F_4|^2 = \frac{g^2}{\kappa_1} + \frac{\gamma_2^2/4}{\kappa_2} - \gamma_2\bar{n}_2\,,$$

$$\text{Re}(F_1)\,\text{Re}(F_4) + \text{Re}(F_2)\,\text{Re}(F_3) = 0\,,$$

$$\text{Re}(F_1)\,\text{Im}(F_4) - \text{Re}(F_2)\,\text{Im}(F_3) = g\left(\frac{\gamma_1/2}{\kappa_1} - \frac{\gamma_2/2}{\kappa_2}\right). \tag{A.25}$$

We only need to find $F_1$ and $F_2$ which in this case are real, since

$$\tilde{K}(\vec{s}) = \frac{1}{2}\,\text{tr}\big[\Gamma_s^{1/2}\mathcal{F} - \mathcal{A}\big] = \frac{\gamma_1 + \gamma_2}{2} + \sqrt{\kappa_1}F_1 + \sqrt{\kappa_2}F_2\,. \tag{A.26}$$

There are several solutions to these equations, however, only of of them provides the correct long-time. This physical solution is found from the branch for which $\text{Re}\,F_3 = \text{Re}\,F_4 = 0$. If one instead would assume that $\text{Re}\,F_3 \neq 0$, it turns out that $\sqrt{\kappa_1}F_1 = -\sqrt{\kappa_2}F_2$, which would make the cumulant generating function a constant, and all cumulants would vanish.

The remaining four equations can be solved by first introducing the new variables

$$R_\pm = \sqrt{\kappa_1}F_1 \pm \sqrt{\kappa_2}F_2\,, \quad \text{and} \quad J_\pm = \sqrt{\kappa_1}\,\text{Im}\,F_3 \pm \sqrt{\kappa_2}\,\text{Im}\,F_4\,. \tag{A.27}$$

From the sum and the difference of the two first equations in Eq. (A.25), and rewriting the other two equations, we then find

$$J_+ = -g\sqrt{\kappa_1\kappa_2}\left(\frac{1}{\kappa_1} + \frac{1}{\kappa_2}\right),$$

$$R_+ J_- - R_- J_+ = -g\sqrt{\kappa_1\kappa_2}\left(\frac{\gamma_1}{\kappa_1} - \frac{\gamma_2}{\kappa_2}\right) \equiv a\,,$$

$$R_+ R_- + J_+ J_- = g^2\left(\frac{\kappa_1}{\kappa_2} - \frac{\kappa_2}{\kappa_1}\right) + \frac{\gamma_1^2 - \gamma_2^2}{4} - (\gamma_1\bar{n}_1\kappa_1 - \gamma_2\bar{n}_2\kappa_2) \equiv b\,,$$

$$\frac{1}{2}\big[R_+^2 + R_-^2 + J_+^2 + J_-^2\big] = g^2\left(\frac{\kappa_1}{\kappa_2} + \frac{\kappa_2}{\kappa_1}\right) + \frac{\gamma_1^2 + \gamma_2^2}{4} - (\gamma_1\bar{n}_1\kappa_1 + \gamma_2\bar{n}_2\kappa_2) \equiv c\,. \tag{A.28}$$

The three variables $R_+$, $R_-$, and $J_-$ determine the cumulant generating function, and one can see that $(R_+^2 + J_+^2)(R_-^2 + J_-^2) = a^2 + b^2$. From the last equation, we find

$$(R_+^2 + J_+^2)^2 - 2c(R_+^2 + J_+^2) + a^2 + b^2 = 0\,, \tag{A.29}$$

which gives

$$R_+ = -\sqrt{-J_+^2 + c + \sqrt{c^2 - a^2 - b^2}}\,. \tag{A.30}$$

By inserting this expression into $\tilde{K}(\vec{s}) = (\gamma_1 + \gamma_2)/2 + R_+$, we arrive at Eq. (56) of the main text.

For the TMS system, the solution follows the same steps, except we begin with the Ansatz

$$\mathcal{F} = \begin{pmatrix} F_1 & 0 & 0 & F_3 \\ 0 & F_1^* & F_3^* & 0 \\ 0 & F_4 & F_2 & 0 \\ F_4^* & 0 & 0 & F_2^* \end{pmatrix}. \tag{A.31}$$

Defining $R_\pm$ and $J_\pm$ as for BS interactions, we obtain the quadratic equation

$$(R_+^2 + J_-^2)^2 - 2\tilde{c}(R_+^2 + J_-^2) + \tilde{a}^2 + \tilde{b}^2 = 0, \tag{A.32}$$

where $\tilde{b} = b(g \to \lambda)$ and $\tilde{c} = c(g \to \lambda)$ are obtained from Eq. (A.28) by redefining $g \to \lambda$, as well as $J_- = \lambda\left(\sqrt{\kappa_1/\kappa_2} - \sqrt{\kappa_1/\kappa_2}\right)$ and $\tilde{a} = \lambda\sqrt{\kappa_1\kappa_2}(2 + \gamma_1/\kappa_1 + \gamma_2/\kappa_2)$. Solving the quadratic equation then leads us to Eq. (59) of the main text.

Finally, we may add a contribution to the cumulant generating function, such that

$$\tilde{K}(\vec{s}) \to \tilde{K}(\vec{s}) + \tilde{K}^{\text{drive}}(\vec{s}). \tag{A.33}$$

For the two cases discussed here, we get from Eq. (45) that

$$\begin{aligned}
\tilde{K}_{\text{BS}}^{\text{drive}}(\vec{s}) &= p_{\text{BS}}[|f_1|^2\kappa_1\gamma_2(\gamma_2 - 4\bar{n}_2\kappa_2) + |f_2|^2\kappa_2\gamma_1(\gamma_1 - 4\bar{n}_1\kappa_1)] \\
&\quad + 4p_{\text{BS}}[g^2(\kappa_2|f_1|^2 + \kappa_1|f_2|^2) + g|f_1f_2|(\gamma_1\kappa_2 - \gamma_2\kappa_1)\sin(\phi_1 - \phi_2)], \\
p_{\text{BS}}^{-1} &= \frac{\gamma_1\gamma_2}{4}(\gamma_1 - 4\bar{n}_1\kappa_1)(\gamma_2 - 4\bar{n}_2\kappa_2) + 2g^2(\gamma_1\gamma_2 - 2\gamma_1\bar{n}_1\kappa_2 - 2\gamma_2\bar{n}_2\kappa_1) + 4g^4,
\end{aligned} \tag{A.34}$$

and

$$\begin{aligned}
\tilde{K}_{\text{TMS}}^{\text{drive}}(\vec{s}) &= p_{\text{TMS}}[|f_1|^2\kappa_1\gamma_2(\gamma_2 - 4\bar{n}_2\kappa_2) + |f_2|^2\kappa_2\gamma_1(\gamma_1 - 4\bar{n}_1\kappa_1)] \\
&\quad + 4p_{\text{TMS}}[\lambda^2(\kappa_2|f_1|^2 + \kappa_1|f_2|^2) - 4\lambda|f_1f_2|(\gamma_2\kappa_1 + \gamma_1\kappa_2)\sin(\phi_1 + \phi_2)], \\
p_{\text{TMS}}^{-1} &= \frac{\gamma_1\gamma_2}{4}(\gamma_1 - 4\bar{n}_1\kappa_1)(\gamma_2 - 4\bar{n}_2\kappa_2) - 2\lambda^2(\gamma_1\gamma_2 + 2\gamma_1\bar{n}_1\kappa_2 + 2\gamma_2\bar{n}_2\kappa_1) + 4\lambda^4,
\end{aligned} \tag{A.35}$$

where the driving amplitudes are given by $|f_{1,2}|$ and their phases by $\phi_{1,2}$.

## A.5 Expansions in counting fields

The photon counting equations are well suited for low-order expansions in the counting fields, which enable easier calculations of the first few cumulants. Here, we focus on the emissions from two cavities and set $\vec{u} = 0$. However, the approach that we follow below works equally well for absorption and also for larger networks.

To begin with, we expand the covariance matrix and the vector of first moments as

$$\Theta_N(t; \vec{s}) = \sum_{n,m=0}^{\infty} \frac{s_1^n s_2^m}{n!m!}\Theta_N^{(nm)}(t), \quad \text{and} \quad \boldsymbol{d}(t; \vec{s}) = \sum_{n,m=0}^{\infty} \frac{s_1^n s_2^m}{n!m!}\boldsymbol{d}^{(nm)}(t). \tag{A.36}$$

For $\vec{s} = 0$, $\Theta_N^{(00)}(t) \equiv \Theta_N(t)$ and $\boldsymbol{d}^{(00)}(t) \equiv \boldsymbol{d}(t)$ are given by Eqs. (9) and (10). Furthermore, the expansion of the matrix $\Gamma_s$ in Eq. (28) for two cavities reads $\Gamma_s = \sum_n \frac{s_1^n}{n!}\Gamma_1 + \sum_m \frac{s_2^m}{m!}\Gamma_2$.

The cumulant generating function depends linearly on $\Gamma_s$. Therefore, it is enough to expand $\Theta_N$ and $\boldsymbol{d}$ up to first order in the counting fields to obtain the cumulant generating function up to second order. Using the photon counting equations (25) as well as $\Theta_N = \Theta - I_{2N}/2$, we find

$$\begin{aligned}
\frac{\mathrm{d}}{\mathrm{d}t}\Theta_N^{(10)} &= \mathcal{A}\Theta_N^{(10)} + \Theta_N^{(10)}\mathcal{A}^\dagger + \Theta_N\Gamma_1\Theta_N, \\
\frac{\mathrm{d}}{\mathrm{d}t}\Theta_N^{(01)} &= \mathcal{A}\Theta_N^{(01)} + \Theta_N^{(01)}\mathcal{A}^\dagger + \Theta_N\Gamma_2\Theta_N, \\
\frac{\mathrm{d}}{\mathrm{d}t}\boldsymbol{d}^{(10)} &= \mathcal{A}\boldsymbol{d}^{(10)} + \Theta_N\Gamma_1\boldsymbol{d}, \\
\frac{\mathrm{d}}{\mathrm{d}t}\boldsymbol{d}^{(01)} &= \mathcal{A}\boldsymbol{d}^{(01)} + \Theta_N\Gamma_2\boldsymbol{d}.
\end{aligned} \tag{A.37}$$

As expected, for $\Gamma_{1,2} = 0$, we have $\Theta_N^{(10)} = \Theta_N^{(01)} = 0$ and $\boldsymbol{d}^{(10)} = \boldsymbol{d}^{(01)} = 0$.

The solutions of these equations are now inserted into Eq. (25c), which gives

$$
\begin{aligned}
\frac{\mathrm{d}}{\mathrm{d}t} K(t;\vec{s}) \simeq \frac{1}{2}\Big[ & s_1\big(\mathrm{tr}\{\Gamma_1 \Theta_N\} + \boldsymbol{d}^\dagger \Gamma_1 \boldsymbol{d}\big) + s_2\big(\mathrm{tr}\{\Gamma_2 \Theta_N\} + \boldsymbol{d}^\dagger \Gamma_2 \boldsymbol{d}\big) \\
& + s_1^2 \Big(\mathrm{tr}\big\{\Gamma_1 \Theta_N^{(10)}\big\} + 2\boldsymbol{d}^\dagger \Gamma_1 \boldsymbol{d}^{(10)} + \frac{1}{2}\mathrm{tr}\{\Gamma_1 \Theta_N\} + \frac{1}{2}\boldsymbol{d}^\dagger \Gamma_1 \boldsymbol{d}\Big) \\
& + s_2^2 \Big(\mathrm{tr}\big\{\Gamma_2 \Theta_N^{(01)}\big\} + 2\boldsymbol{d}^\dagger \Gamma_2 \boldsymbol{d}^{(01)} + \frac{1}{2}\mathrm{tr}\{\Gamma_2 \Theta_N\} + \frac{1}{2}\boldsymbol{d}^\dagger \Gamma_2 \boldsymbol{d}\Big) \\
& + s_1 s_2 \Big(\mathrm{tr}\big\{\Gamma_1 \Theta_N^{(01)} + \Gamma_2 \Theta_N^{(10)}\big\} + 2\boldsymbol{d}^\dagger \Gamma_2 \boldsymbol{d}^{(10)} + 2\boldsymbol{d}^\dagger \Gamma_1 \boldsymbol{d}^{(01)}\Big)\Big],
\end{aligned}
\tag{A.38}
$$

up to the second order in the counting fields. The cumulants can then be read from the coefficients of the different powers of $s_1^n s_2^m$. The integration of this equation is straightforward with the initial condition $K(0,\vec{s}) = 0$. To evaluate the cumulants at long times, one can substitute $\frac{\mathrm{d}}{\mathrm{d}t} K(t;\vec{s})$ by $K(t;\vec{s})/t$ and replace the expansion parameters by their steady-state values.

The expansion also allows for establishing a connection between the short and long-time statistics in the steady state, which is often presented only for a single system [65]. In the long-time limit, the ratios of the second cumulants over the average values, also known as the Fano factors, can be related to the second-order coherence functions as shown in Eq. (47) This equation can be derived as follows: First, one notes that Eqs. (A.37,A.38) generalize straightforwardly to any two cavity modes in a cavity network. In the steady state, we have $\Theta_N(t) = \Theta_{N,0}$ and $\boldsymbol{d}(t) = \boldsymbol{d}_0$. Then, one can integrate the second-order coherence functions $g_{jk}^{(2)}$ using the result derived in Eq. (42). The connection is established by the integrals

$$
\Theta_N^{(j)} = \int_0^\infty \mathrm{d}\tau\, e^{\mathcal{A}\tau} \Theta_{N,0} \Gamma_j \Theta_{N,0} e^{\mathcal{A}^\dagger \tau}, \quad \text{and} \quad \boldsymbol{d}^{(j)} = \int_0^\infty \mathrm{d}\tau\, e^{\mathcal{A}\tau} \Theta_{N,0} \Gamma_j, \tag{A.39}
$$

which solve the Lyapunov and vector equations, similar to Eq. (A.37),

$$
\mathcal{A}\Theta_N^{(j)} + \Theta_N^{(j)} \mathcal{A}^\dagger + \Theta_{N,0} \Gamma_j \Theta_{N,0} = 0, \quad \mathcal{A}\boldsymbol{d}^{(j)} + \Theta_{N,0}\Gamma_j \boldsymbol{d}_0 = 0. \tag{A.40}
$$

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
