# Peer review of "Photon counting statistics in Gaussian bosonic networks"

_SciPost Physics, doi:SciPost Phys. 18, 116 (2025)_

## Round 1 · Referee Report · Anonymous (Referee 1) · 2024-9-27

Strengths

1- the reader can obtain an in-depth understanding of photon counting and the method 2- clear potential for follow-up work in a variety of directions 3- well-written and illustrated

Weaknesses

1- missing connection to an alternative approach

Report

The authors present a method to obtain the photon counting statistics of Gaussian bosonic networks where every bosonic subsystem is embedded in an individual thermal environment. The time evolution of the full network is described by a Lindblad master equation that is at most quadratic in the bosonic ladder operators. The main method the authors employ are Gaussian states which are naturally applicable (the Liouvillian is quadratic). The authors introduce necessary concepts to understand the connection between the Lindblad equation, probabilties for photon emission and absorption, and the (cumulant) generating function of the photon current. The cumulant generating function can be obtained by solving three coupled differential equations. Additionally, the two-time correlators ‘waiting time distribution’ and the ‘second-order coherence’ are discussed; both quantities describe the correlation of the radiation. After explaining how to obtain the photon counting statistics at long times, the authors finish by applying their method to three examples. There, they also discuss a potential entanglement witness that can be obtained from the photon counting.

The paper is well-written and given the usefulness of the method, the length of the manuscript is definitely justified. All key ingredients to understand the subject are stated in the main text. Further, additional information and details can be found in the appendices such that every reader can obtain an in-depth understanding of the topic.

In the introduction, the authors state that there does not yet exist a method to obtain the counting statistics of Gaussian bosonic networks. Regarding this, the authors need to be aware of the concept of ‘third quantization’ which also works on quadratic Liouvillians. Relevant references would be, e.g., [arXiv:1007.2921] (Lyapunov equation for covariance matrix), [arXiv:2304.02367] (generating functions), [arXiv:2302.14047] (connection to Wigner function). What is the connection between the two methods? What are respective advantages and disadvantages? I suggest to add an appendix to discuss this (see points 1 and 2 in the requested changes).

This (and other points) have to be appropriately addressed in the revised manuscript.

Requested changes

1- What is the direct connection between Gaussian states and third quantization? I believe that it is related to the fact that the algebraic Riccati equation can be solved by an eigenvalue decomposition (Wikipedia).

2- What are advantages and disadvantages of both methods? Is there something that can be calculated in one framework easier than in the other? Include an appendix to discuss point 1 and 2.

3- For the TMS-system, or nondegenerate parametric oscillator, the full counting statistics have already been obtained in the past, e.g., in Phys. Rev. A 46, 395. Please include relevant references and compare the results.

4- After the submission of the paper, advancements in the detection of single microwave photons have been made, e.g, [Phys. Rev. Lett. 133, 076302]. Maybe update the list of references regarding this point.

  1. How is the waiting time distribution impacted by a finite detection efficiency? Maybe include a numerical simulation for such a case.

6- While it is mentioned and cited, the authors might consider to explicitly include the connection between the second-order coherence and the Fano factor to provide a more complete picture and draw a nice connection between the short- and long-time statistics.

7- I did not find a definition of $\Theta_{N,0} $. Is $\Theta_{N,0} =\Theta_0 – I_{2N}/2$ (page 11) where $\Theta_0$ is obtained from Eq.10?

8- While the Hamiltonian of the system can be deduced by the matrix $\mathcal{A}$ it might be useful for the reader when the Hamiltonians are written down in section 4.

Recommendation

Ask for major revision

  • validity: high
  • significance: high
  • originality: good
  • clarity: high
  • formatting: excellent
  • grammar: excellent

Author:  Kalle Kansanen  on 2025-01-30  [id 5167]

(in reply to Report 1 on 2024-09-27)

The authors present a method to obtain the photon counting statistics of Gaussian bosonic networks where every bosonic subsystem is embedded in an individual thermal environment. The time evolution of the full network is described by a Lindblad master equation that is at most quadratic in the bosonic ladder operators. The main method the authors employ are Gaussian states which are naturally applicable (the Liouvillian is quadratic). The authors introduce necessary concepts to understand the connection between the Lindblad equation, probabilities for photon emission and absorption, and the (cumulant) generating function of the photon current. The cumulant generating function can be obtained by solving three coupled differential equations. Additionally, the two-time correlators ‘waiting time distribution’ and the ‘second-order coherence’ are discussed; both quantities describe the correlation of the radiation. After explaining how to obtain the photon counting statistics at long times, the authors finish by applying their method to three examples. There, they also discuss a potential entanglement witness that can be obtained from the photon counting.

The paper is well-written and given the usefulness of the method, the length of the manuscript is definitely justified. All key ingredients to understand the subject are stated in the main text. Further, additional information and details can be found in the appendices such that every reader can obtain an in-depth understanding of the topic.

Our response: Thank you for the careful reading of our manuscript and for the detailed and accurate report. We have implemented most of the changes that you suggest and below we provide a point-to-point response to your comments. Note that when we discuss our manuscript, the equation numbers refer to the revised manuscript.

In the introduction, the authors state that there does not yet exist a method to obtain the counting statistics of Gaussian bosonic networks. Regarding this, the authors need to be aware of the concept of ‘third quantization’ which also works on quadratic Liouvillians. Relevant references would be, e.g., [arXiv:1007.2921] (Lyapunov equation for covariance matrix), [arXiv:2304.02367] (generating functions), [arXiv:2302.14047] (connection to Wigner function).

Our response: Thank you for this reminder, and we are in fact well aware of third quantization. Thus, forgetting to mention third quantization in the introduction was clearly a blunder on our side, and we have now updated the introduction, so that methods based on third quantization are explicitly discussed together with proper citations of relevant references.

Corresponding change: We have revised the introduction and now discuss related works on third quantization.

1- What is the direct connection between Gaussian states and third quantization? I believe that it is related to the fact that the algebraic Riccati equation can be solved by an eigenvalue decomposition (Wikipedia).

Our response: That is a very good question, and it would be interesting to understand the direct connection between the two approaches. Clearly, third quantization and phase space calculations must be connected since the underlying quantum master equation is the same. However, the exact mathematical connection may not be easy to establish, and we are not aware of it.

To address the question, we note that the Riccati equation can be transformed into a linear equation using a lemma by Radon [Eq. (2.1) in the book by H. Abou-Kandil et al.]. By defining two auxiliary matrices $Q(t)$ and $P(t)$, such that $\Theta_N(t) = P(t)Q(t)^{-1}$ and $Q(0) = I_{2N}$, the Riccati equation in the normal-ordered representation [Eq. (55)] takes the form

$$ \begin{pmatrix} \dot{Q}(t) \\ \dot{P}(t) \end{pmatrix} = \begin{pmatrix} -\mathcal{A}^\dagger & - \Gamma_s \\ \mathcal{B} & \mathcal{A} \end{pmatrix} \begin{pmatrix} Q(t) \\ P(t) \end{pmatrix} = \mathcal{H}_S \begin{pmatrix} Q(t) \\ P(t) \end{pmatrix}. $$
Here, the symplectic Hamiltonian $\mathcal{H}_S$ appears to be closely related to the Liouvillian in the third quantization formalism [Eq. (19) in arXiv:2304.02367]. As you mention, the matrix $\mathcal{H}_S$ determines the solution of the steady state (which is the algebraic Riccati equation), but it also governs the dynamics. However, we are not sure about the relationship between the matrices $Q, P$ and the bosonic superoperators, and that would be needed to better connect the two approaches. Thus, to bridge the two methods, further investigations are needed, and we will leave that for future work.

2- What are advantages and disadvantages of both methods? Is there something that can be calculated in one framework easier than in the other? Include an appendix to discuss point 1 and 2.

Our response: Let us start with the advantages of our method. The phase-space approach is well established in quantum optics and related fields, with different representations ($P$-, $Q$-, and Wigner distributions) corresponding to different physical situations. We are able to describe the photon counting statistics with the first two moments of Gaussian states, and the resulting equations for the covariance matrix and first moments are well-known matrix differential equations. Furthermore, we can connect these equations to time-dependent quantities such as the waiting time distribution with an additional set of differential equations. These equations are simple to solve numerically and also allow for analytic solutions in some cases. On the other hand, the third quantization appears to provide a framework that is more easily generalized and connected to other approaches. For small and simple systems, presumably most quantities can be derived with a similar effort, but it is not clear to us how one would approach the calculation of the WTDs within the third quantization scheme.

Corresponding change: We now shortly discuss the points above at the end of App. A.2.

3- For the TMS-system, or nondegenerate parametric oscillator, the full counting statistics have already been obtained in the past, e.g., in [Phys. Rev. A 46, 395] (not cited). Please include relevant references and compare the results.

Our response: The article by R. Vyas is indeed relevant and interesting, as it shows that the photon counting statistics of joint quantities (summed intensities or amplitudes from the output of the two cavities) can be fully determined for the TMS system (nondegenerate parametric amplifier). A point of direct comparison can be found in the counting statistics of the summed intensities in the long-time limit: One can confirm that our result for the cumulant generating function in Eq. (59) corresponds exactly to Eq. (27) of Vyas's article after a set of substitutions to align the definitions. If the parameters of Vyas's PRA are denoted by subscript $V$ and one sets $\bar{n}_{1,2} = 0$, $s_1 = s_2 = s$ and $\gamma_1 = \gamma_2 = \gamma$ in our Eq. (59), the substitutions are $s_V = 1 - e^s, \gamma_V = \gamma/2, |\kappa \epsilon|_V = \lambda$, and finally $\tilde{K}(s) =\lim_{T\rightarrow\infty} \frac{1}{T}\ln(G_{sum}(s_V,T))_V$.

Corresponding change: We now mention these connections in the sentences following Eq. (61).

4- After the submission of the paper, advancements in the detection of single microwave photons have been made, e.g, [Phys. Rev. Lett. 133, 076302]. Maybe update the list of references regarding this point.

Corresponding change: We have added the reference to the introduction, as well as the following recent articles: Phys. Rev. Applied 21, 014043; PRX Quantum 5, 020342; Phys. Rev. Lett. 133, 217001; Phys. Rev. X 14, 011011; Phys. Rev. X 11, 011027.

5- How is the waiting time distribution impacted by a finite detection efficiency? Maybe include a numerical simulation for such a case.

Our response: As discussed below Eq. (19), a finite detection efficiency can be taken into account by including a factor $\eta_j$ together with the emission counting field terms in the generalized quantum master equation. In essence, the $\eta_j$ factors modify the $\Gamma_j$ matrices, which in turn are the block diagonal elements of the $\Gamma_s$ matrix. (The underlying argument is that we can model finite detection efficiency with multiple environments to each cavity, while only photons emitted into one of them, acting as the detector, are counted.)

As such, the effect of finite detection efficiency can easily be calculated numerically. Analytically, we can evaluate the short- and long-time behaviors. Following the prescription of setting $\Gamma_j \rightarrow \eta_j \Gamma_j$, we can see that the WTD at short times behaves linearly with respect to the detector efficiency, $W_{jk}(0) \rightarrow \eta_k W_{jk}(0)$. At long times, the exponential decay rate of the WTD is nonlinear in $\eta_k$, but finite detection efficiency always diminishes this rate. Therefore, a detector inefficiency will generally lead to a flattened distribution with larger expected times between photon events, which is a sensible result.

Corresponding change: We have included two more panels in Fig. 3 to show the effect of a finite detection efficiency on the WTDs. We have also added a paragraph discussing the details in Sect. 4.1 between Eqs. (54) and (55), as well as a more general discussion to the end of Sect. 3.3.

6- While it is mentioned and cited, the authors might consider to explicitly include the connection between the second-order coherence and the Fano factor to provide a more complete picture and draw a nice connection between the short- and long-time statistics.

Our response: We agree with your point, and we have revised the manuscript accordingly.

Corresponding change: In the last paragraph of Section 3.4, we now present how the Fano factors can be calculated from the second-coherence functions, while the end of Appendix A.5 explains the derivation.

7- I did not find a definition of $\Theta_{N,0}$. Is it $\Theta_{N,0} = \Theta_0 - I_{2N}/2$ (page 11) where $\Theta_0$ is obtained from Eq.10?

Our response: Yes, you are right.

Corresponding change: Below Eq. (41), we now define $\Theta_{N,0}$ explicitly.

8- While the Hamiltonian of the system can be deduced by the matrix $\mathcal{A}$ it might be useful for the reader when the Hamiltonians are written down in section 4.

Our response: We agree that this suggestion would help the reader.

Corresponding change: We have now added the relevant Hamiltonians at the beginning of Sect. 4.1.

---

## Round 1 · Referee Report · Anonymous (Referee 2) · 2024-10-17

Strengths

1) The paper is overall well written and is also pedagogical. 2) The overview of the field, i.e. list of references, is satisfactory. 3) The steps of the theoretical analysis and the flow of the calculations can be followed very straightforwardly. An useful compact notation (matrix form) is used.

Weaknesses

Examples of the developed method are provided exclusively for the simplest cases, specifically for two or three coupled bosonic modes. Some of the quantities analyzed, such as correlation functions and entanglement for the case of two coupled modes, could, in principle, be derived without employing the proposed method.

Report

The authors investigate the full counting statistics of N coupled bosonic modes, considered a many-body open quantum system. Two of the authors have expanded upon their earlier work (Ref. [35]), extending the analysis from a single resonator to the case of N coupled resonators. The work is intriguing and, in my opinion, warrants publication, provided that the authors satisfactorily address my questions and properly resolve the concerns outlined below.

1) The model Hamiltonian in Eq. (1) is presented in the rotating frame of the driven resonators. The authors state that the resonators "are all driven at the frequency $\omega_d$" which is not accurate. The squeezed term appearing in Eq. 1 ($\sim {\hat{a}^{\dagger}}^2_i$) originates from the parametric pumping at $2 \omega_d$. If this were not the case, after transforming to the rotating frame, the parametric pump would manifest as fast-rotating and, by applying the rotating wave approximation (RWA), it would need to be disregarded. Therefore, the authors should revise the text accordingly.

2) Since the system is purely linear, the parametric pumping at the drive frequency $2\omega_d$ with $\omega_d \sim \omega_i$ can yield divergent parametric oscillations (self-induced oscillations) unless the system is sufficiently detuned from the region of such instability (Arnold's tongue) and/or the strength of the parametric drive $r_j$ is significantly smaller than the damping $\gamma_i$ (numerical coefficients may vary based on the model's notation). Could the authors provide a comment on this point?

3) There is another implicit assumption in their model that has not been explicitly mentioned: The frequencies of the resonators must be nearly equal, within a specified range determined by the resonator losses, i.e., $\gamma_i$, namel $\omega_i \sim \omega_d$. If this condition is not met, one resonator will be driven out of resonance, which effectively means it is not driven at all, particularly if the detuning is much larger than the linewidth of its response function. Technically, this implies that, by applying the rotating wave approximation (RWA), the coherent drive $f_i$ for the resonator with $|\omega_d - \omega_i| \gg \gamma_i$ will appear as fast oscillating and would need to be disregarded in the Hamiltonian.

4) The authors should emphasize that the Lindblad equation in Eq. (3) describes a drive-dissipative open quantum system, specifically a system out of thermal equilibrium. This is linked to the observation that the operators appearing in the dissipator in Eq. (3) are not associated with the "true" eigenstates of the quadratic Hamiltonian that describes the system, therefore the system cannot reach the thermal equilibrium. In this regard, in addition to citing Ref. [42], which pertains only to equilibrium conditions, the authors may consider citing, for example: (i) [https://arxiv.org/abs/2409.10300] and/or (ii) [https://arxiv.org/abs/2407.16855].

5) Related to the previous issue, the authors discuss the validity of the Lindblad equation that includes the dissipator operators identically to a system of non-coupled/non-interacting resonators, which is also known as the "additive approximation". They assert that this approximation is valid for 'weak coupling in the network compared to the dissipation rates $\gamma_i$', namely $(g_{jk}, \lambda_{jk})\ll\gamma_i$. However, the authors present results with $g = \gamma$, which contradicts their assertion. However, the regime $(g, \lambda) \ll \gamma$ is fully dissipative and is opposite to the strong coupling regime $(g, \lambda) \gg \gamma$ , where the coherent interaction between the different modes is larger than the dissipative rate, potentially leading to interesting effects (e.g., entanglement, etc.). I am not sure whether the condition provided by the authors is the right one. For instance, in circuit QED (one qubit coupled to a resonator), the 'additive approximation' is frequently employed even in the "strong coupling regime", and it functions effectively. The system proposed here seems analogous: if we consider a transmon as a qubit (i.e., a nonlinear resonator) coupled to a cavity (a second resonator), I see no fundamental difference between the two systems. The "additive approximation" surely breaks down in the Ultra-Strong Coupling (USC) regime in which we have $(g,\lambda\sim \omega_i)$, see a recent paper Phys. Rev. Lett. vol. 132, p.106902. Although it has been well-established that decay operators derived from an uncoupled system can result in unphysical effects when applied to a coupled system [J. Phys. A 6, 1552 (1973)], their use is frequently considered a valid approximation. Could the authors analyze better this point?

6) The authors have developed a powerful theoretical framework, and it would be valuable to see how this approach can be applied. I understand that the first two examples analyzed (two coupled modes and entanglement) are pedagogical in nature, intended to illustrate the implementation of the method. The last example, involving the three-mode circulator, is particularly intriguing due to its broken time-reversal symmetry. In the final part of this section, the authors consider the case without the coherent drive. In their calculations, they assume (or fix) the net current flowing between cavities 1 and 2, and then they present the probability distribution of the current. However, why is it not possible to 'derive' the net current flowing through the system and demonstrate the broken time-reversal symmetry instead of assuming it from the outset? The proposed method should be capable of directly computing the current. Could the authors provide a clearer explanation of this issue?

Finally, I would like to comment on the authors' last statement at the end of the paper: 'It may be possible to extend our framework to non-Gaussian systems with Kerr or other nonlinearities.' This seems improbable a priori. The ansatz of a Gaussian state is heavily utilized, particularly for transitioning from the Lindblad equation to time-dependent equations for the correlators, averages, and shifts. In principle, non-Gaussian states could lead to a hierarchy of equations for higher-order cumulants. Could the authors provide better insight into how it might be possible to extend the method to accommodate non-Gaussian systems?

Requested changes

  • After Eq. (17), there is an abrupt discontinuity in the theoretical derivation. Specifically, the quantity $\rho(\vec{n}, \vec{m}, t)$ is neither defined by a formula nor through an equation. Although the authors refer to Refs. [54, 55], for the sake of completeness, it would be helpful to include a formula or equation here to define $\rho(\vec{n}, \vec{m}, t)$.

  • In Eq.(14) it appears a sum over the modes $\sum_{j=1}^N$ which I don't understand. The quantity $P(t;\vec{n},\vec{m})$ already includes all the modes so why do we need to sum? or maybe I missed something?

Recommendation

Ask for major revision

  • validity: high
  • significance: good
  • originality: good
  • clarity: high
  • formatting: excellent
  • grammar: excellent

Author:  Kalle Kansanen  on 2025-01-30  [id 5168]

(in reply to Report 2 on 2024-10-17)

Warnings issued while processing user-supplied markup:

  • Coercing language: Markdown
  • Inconsistency: Markdown and reStructuredText syntaxes are mixed. Markdown will be used.
    Add "#coerce:reST" or "#coerce:plain" as the first line of your text to force reStructuredText or no markup.
    You may also contact the helpdesk if the formatting is incorrect and you are unable to edit your text.

The authors investigate the full counting statistics of N coupled bosonic modes, considered a many-body open quantum system. Two of the authors have expanded upon their earlier work (Ref. [35]), extending the analysis from a single resonator to the case of N coupled resonators. The work is intriguing and, in my opinion, warrants publication, provided that the authors satisfactorily address my questions and properly resolve the concerns outlined below.

Our response: Thank you for the careful reading of our manuscript and the detailed report, which has helped us clarify several issues of relevance to circuit QED. Also, thank you for the positive feedback, and we are happy to learn that you seem inclined to recommend publication of our work. As a very minor comment, let us briefly mention that we were all involved in Ref. [35], which is Ref. [42] in the revised manuscript.

1) The model Hamiltonian in Eq. (1) is presented in the rotating frame of the driven resonators. The authors state that the resonators "are all driven at the frequency $\omega_d$ which is not accurate. The squeezed term appearing in Eq. 1 (${a_i^\dagger}^2$) originates from the parametric pumping at $2\omega_d$. If this were not the case, after transforming to the rotating frame, the parametric pump would manifest as fast-rotating and, by applying the rotating wave approximation (RWA), it would need to be disregarded. Therefore, the authors should revise the text accordingly.

Our response: Thank you for this comment, and you are right about the physical origin of the single-mode squeezing terms. While we attempt to separate coherent driving from parametric driving, it is true that the sentence in the beginning of Sect. 2 was not accurately worded.

Corresponding change: In the first paragraph of Sect. 2, we have added the word "coherently" to the sentence "The cavities ... are all coherently driven at the frequency $\omega_D$." Later in the same paragraph, we then explain the origin of the single-mode interaction as "arising from parametrically driving the cavity with a frequency close to $2\omega_j$", and in the last sentence we have added "coherent and parametric" to clearly separate these forms of driving.

2) Since the system is purely linear, the parametric pumping at the drive frequency $2 \omega_d$ with $\omega_d \sim \omega_i$ can yield divergent parametric oscillations (self-induced oscillations) unless the system is sufficiently detuned from the region of such instability (Arnold's tongue) and/or the strength of the parametric drive $r_j$ is significantly smaller than the damping $\gamma_i$ (numerical coefficients may vary based on the model's notation). Could the authors provide a comment on this point?

Our response: That is correct, and we mention it in a somewhat roundabout way: In the regime of self-induced oscillations, the dynamical matrix $\mathcal{A}$ has eigenvalues with positive real parts and, as expected, there is no stationary solution for the density matrix. Here, we choose to focus on stable systems without such oscillations, and we mention that just below Eq. (13).

Corresponding change: We now clarify this point below Eq. (13) with the sentence "This limits the possible values of the parameters $r_j$ and $\lambda_{jk}$ corresponding to single-mode and two-mode squeezing interactions, respectively."

3) There is another implicit assumption in their model that has not been explicitly mentioned: The frequencies of the resonators must be nearly equal, within a specified range determined by the resonator losses, i.e., $\gamma_i$, namely $\omega_i \sim \omega_d$. If this condition is not met, one resonator will be driven out of resonance, which effectively means it is not driven at all, particularly if the detuning is much larger than the linewidth of its response function. Technically, this implies that, by applying the rotating wave approximation (RWA), the coherent drive $f_i$ for the resonator with $|\omega_d - \omega_i| \gg \gamma_i$ will appear as fast oscillating and would need to be disregarded in the Hamiltonian.

Our response: It is true that if that there are several cavities that are coherently driven, they have to share the same eigenfrequency which is a consequence of the cavity network setup in Sect. 2. However, if the cavities are not driven, there are no restrictions on the eigenfrequencies. Different eigenfrequencies then show up only in the diagonal of the dynamical matrix $\mathcal{A}$, and one can, for instance, solve the photon-counting statistics in the long-time limit as discussed in Sect. 4 and App. A.4.

Corresponding change: We have added a clarifying sentence at the end of the first paragraph in Sect. 2: "This choice implies that the cavity eigenfrequencies have to be the same in the presence of driving."

4) The authors should emphasize that the Lindblad equation in Eq. (3) describes a drive-dissipative open quantum system, specifically a system out of thermal equilibrium. This is linked to the observation that the operators appearing in the dissipator in Eq. (3) are not associated with the "true" eigenstates of the quadratic Hamiltonian that describes the system, therefore the system cannot reach the thermal equilibrium. In this regard, in addition to citing Ref. [42], which pertains only to equilibrium conditions, the authors may consider citing, for example: (i) [https://arxiv.org/abs/2409.10300] and/or (ii) [https://arxiv.org/abs/2407.16855].

Our response: Regarding the quantum master equation in Eq. (3), you are correct that it describes a dissipative cavity network which, when driven, is out of equilibrium. We also thank you for the references, which we have included in the manuscript. We discuss the question of dissipators in the response to the next point. However, we would like to stress that our general approach for finding the photon-counting statistics is applicable to any Gaussian bosonic networks, whether it is a driven-dissipative system or a system in thermal equilibrium, and that we have applications to both.

Corresponding change: We now introduce Eq. (3) with a sentence starting with "The cavity network is therefore a dissipative and potentially driven system, whose density matrix [...]" in which we also cite arXiv:2409.10300 and arXiv:2407.16855.

5) Related to the previous issue, the authors discuss the validity of the Lindblad equation that includes the dissipator operators identically to a system of non-coupled/non-interacting resonators, which is also known as the "additive approximation". They assert that this approximation is valid for 'weak coupling in the network compared to the dissipation rates $\gamma_i$', namely $(g_{jk}, \lambda_{jk}) \gg \gamma_i$. However, the authors present results with $g = \gamma$, which contradicts their assertion. [...]

Our response: We agree with your comment about the additive approximation. The sentence of the manuscript that you mention is of course true, but it represents a case in which one can be absolutely sure that the local description of dissipation is valid. However, in practice, the coupling strength should indeed be compared to the eigenfrequencies of the system, as this ratio controls the difference between local and global dissipators. Whenever this ratio is small, which we assume, the local dissipators and the additive approximation tend to work well as confirmed by many experiments on cQED. Even though some thermodynamic inconsistencies may arise, these do not affect our observables in any significant way.

Corresponding change: In the paragraph following Eq. (4), we have added the sentences "In practice, local dissipators function well up until the ultrastrong coupling regime, where the coupling strengths become comparable to the cavity eigenfrequencies [8]. That is, the following results are applicable to systems where the cavity eigenfrequencies are larger than the couplings and the dissipation rates."

6) [...] In the final part of this section, the authors consider the case without the coherent drive. In their calculations, they assume (or fix) the net current flowing between cavities 1 and 2, and then they present the probability distribution of the current. However, why is it not possible to 'derive' the net current flowing through the system and demonstrate the broken time-reversal symmetry instead of assuming it from the outset?

Our response: Let us here clarify that we consider the probability distribution $P(J_1,J_2,J_3)$ for the net photon currents, $J_i$, running into each of the three cavities. In Fig. 7(c), we then show a certain marginal [essentially $\int dJ_3 P(J, -J, J_3)$] of this distribution. As such, we do not fix any currents but rather we obtain the full distribution of the currents from which the averages can be calculated. In thermal equilibrium, the average currents vanish as we mention in the caption of Fig. 7.

Corresponding change: We chanced the first sentence of the relevant part of the manuscript, the paragraph containing Eqs. (79) and (80), to mention "the distribution of net photon currents" instead of just "net photon currents".

Finally, I would like to comment on the authors' last statement at the end of the paper: 'It may be possible to extend our framework to non-Gaussian systems with Kerr or other nonlinearities.' This seems improbable a priori. The ansatz of a Gaussian state is heavily utilized, particularly for transitioning from the Lindblad equation to time-dependent equations for the correlators, averages, and shifts. In principle, non-Gaussian states could lead to a hierarchy of equations for higher-order cumulants. Could the authors provide better insight into how it might be possible to extend the method to accommodate non-Gaussian systems?

Our response: This part is of course rather speculative, but our idea is that it might be possible to use the photon counting equations for the characteristic function or the Wigner function. The approach would then follow the same blueprint: If there is a system whose typical quantum states have a simple parametrization in terms of the Wigner function (e.g., Schrödinger cat states as in one of our references), the resulting equations for the photon counting statistics (similar to the Riccati equation for Gaussian states) could perhaps be solved. In that case, we would not rely on the assumptions of a Gaussian state, and we would thereby go beyond the scope of the current manuscript.

After Eq. (17), there is an abrupt discontinuity in the theoretical derivation. Specifically, the quantity $\rho(\vec{n}, \vec{m}, t)$ is neither defined by a formula nor through an equation. Although the authors refer to Refs. [54, 55], for the sake of completeness, it would be helpful to include a formula or equation here to define $\rho(\vec{n}, \vec{m}, t)$.

Our response: While we would like not to repeat the derivation from the references for the sake of brevity, we have tried to clarify the wording and added the decomposition equation $\rho(t) = \sum_{i=1}^N\sum_{n_i}\sum_{m_i} \rho(t; \vec{n}, \vec{m})$ to clarify the meaning of resolving the density matrix based on the emission and absorption of photons.

Corresponding change: In the text between Eqs. (17) and (18), we have added the sentence "These photon-resolved density matrices decompose the network's density matrix as $\hat\rho(t) = \sum_{j=1}^N\sum_{n_j}\sum_{m_j} \hat\rho(t; \vec{n}, \vec{m})$."

In Eq.(14) it appears a sum over the modes $\sum_{j=1}^N$ which I don't understand. The quantity $P(t; \vec{n}, \vec{m})$ already includes all the modes, so why do we need to sum? or maybe I missed something?

Our response: Equation (14) can be written in a more compact form as

$$ M(t;\vec{s}, \vec{u}) = \sum_{\vec{n}} \sum_{\vec{m}} P(t;\vec{n}, \vec{m}) \exp(\vec{n}\cdot\vec{s} + \vec{m}\cdot\vec{u}), $$
and then there is no explicit sum over the modes. By contrast, with Eq. (14) we explicitly sum over each element in the vectors $\vec{n}$ and $\vec{m}$ as
$$ M(t;\vec{s}, \vec{u}) = \sum_{j=1}^N\sum_{n_j = 0}^\infty \sum_{m_j = 0}^\infty P(t;\vec{n}, \vec{m}) \exp(\vec{n}\cdot\vec{s} + \vec{m}\cdot\vec{u}), $$
and that is why the sum over the modes appears. Importantly, the two expressions are completely equivalent, and one of them is just more compact.

---

## Round 2 · Referee Report · Anonymous (Referee 1) · 2025-3-5

Report

The authors have addressed my comments properly. I recommend that the paper is accepted.

Recommendation

Publish (meets expectations and criteria for this Journal)

---

## Round 2 · List of Changes

- We have revised the article in many places and updated the references based on the two reports. Please, see our point-to-point responses, which also describe the corresponding changes.
- In addition, we have corrected a few minor errors as listed below. (The equation numbers below refer to the revised manuscript.)
- The vector of first moments was mistakenly defined with an extra minus sign in Eq. (6). We have corrected this throughout the text.
- We have corrected minor errors in Eqs. (33), (34), (54), and (56).
- In Fig. 5, the covariance curve for the BS system contained an extraneous prefactor of 0.0025, which has been corrected.
- We have clarified that the expressions in Appendix A.4 refer to the normal-ordered Riccati equation in the first paragraph of the corresponding section.

---

## Editorial Decision

published